# Control of primary metabolism by a virulence regulatory network promotes robustness in a plant pathogen

Rémi Peyraud[1], Ludovic Cottret [1], Lucas Marmiesse[1] & Stéphane Genin [1]

Robustness is a key system-level property of living organisms to maintain their functions while tolerating perturbations. We investigate here how a regulatory network controlling multiple virulence factors impacts phenotypic robustness of a bacterial plant pathogen. We reconstruct a cell-scale model of *Ralstonia solanacearum* connecting a genome-scale metabolic network, a virulence macromolecule network, and a virulence regulatory network, which includes 63 regulatory components. We develop in silico methods to quantify phenotypic robustness under a broad set of conditions in high-throughput simulation analyses. This approach reveals that the virulence regulatory network exerts a control of the primary metabolism to promote robustness upon infection. The virulence regulatory network plugs into the primary metabolism mainly through the control of genes likely acquired via horizontal gene transfer, which results in a functional overlay with ancestral genes. These results support the view that robustness may be a selected trait that promotes pathogenic fitness upon infection.

[1] LIPM, Université de Toulouse, INRA, CNRS, 31326 Castanet-Tolosan, France. Correspondence and requests for materials should be addressed to R.P. (email: remi.peyraud@inra.fr) or to S.G. (email: stephane.genin@inra.fr)

Living organisms have to face changing environments and are subject to accumulation of molecular damages during their growth and development. Therefore, they need to exhibit a certain degree of robustness to sustain their resilience in response to these perturbations. Robustness is commonly defined as the capacity of a biological system to maintain its function(s) despite environmental or internal perturbations; internal perturbations referring here to deleterious mutations or stochastic gene expression[1–3]. Depending on the biological function to be maintained and the kinds of perturbations considered, robustness can arise from various types of molecular mechanisms (Fig. 1). Among them are (i) versatility, i.e., the ability of the system to collect its needs (nutrition, information) from different sources in the environment[4], (ii) functional redundancy, which includes genetic redundancy[5,6] and fail-safe alternatives (alternative metabolic or regulatory pathways), and (iii) the system control, i.e., the capacity of the system to sense and compensate appropriately the undergone perturbation to maintain its homeostasis. This variety of mechanisms and their tight imbrications make robustness challenging to study. Nevertheless, as these mechanisms carry a buffering capacity, studying robustness is key to understand accommodation capacity as well as evolvability of biological systems[1,7]. For instance, genetic redundancy is recognized as a main source of robustness with respect to genetic (internal) perturbation[8]. The adaptive backup capacity provided by genetic redundancy was also proposed to be a transient byproduct during the functional divergence of gene duplicates[9], which eventually could promote evolvability[1].

Phenotypic robustness of living organisms is recognized as a network-scale property[1]. Robustness provided by the metabolic network has been extensively studied[10–13]. One challenge in studying properties of complex networks like metabolism, which harbors a high number of pathway combinations, remains in the computational capacity to explore the large solution space of the system. The development of flux balance analysis (FBA) methods[14], which rely on constraint-based modeling and steady-state assumption has provided the capacity to investigate the sources of robustness at genome-scale level[11,15–17]. Modeling and experimental approaches revealed that robustness of metabolic networks challenged with genetic perturbations relies on pathway redundancy or genetic redundancy, but their extent can vary depending of the organisms[10,15].

Robustness provided by regulatory networks received attention because it determines broad adjustments of cellular processes through the genetic regulation of metabolism, signal transduction, and cell differentiation processes[1,18–20]. Various types of regulation via feed-back loops within the metabolism have been discovered, most of them being aimed to maintain consistency between the biosynthesis of metabolites and their consumption upon perturbation[21–23]. There is evidence that an increase of fitness in various organisms was obtained through modifications of the regulatory network structure[24–27]. Indeed, inappropriate

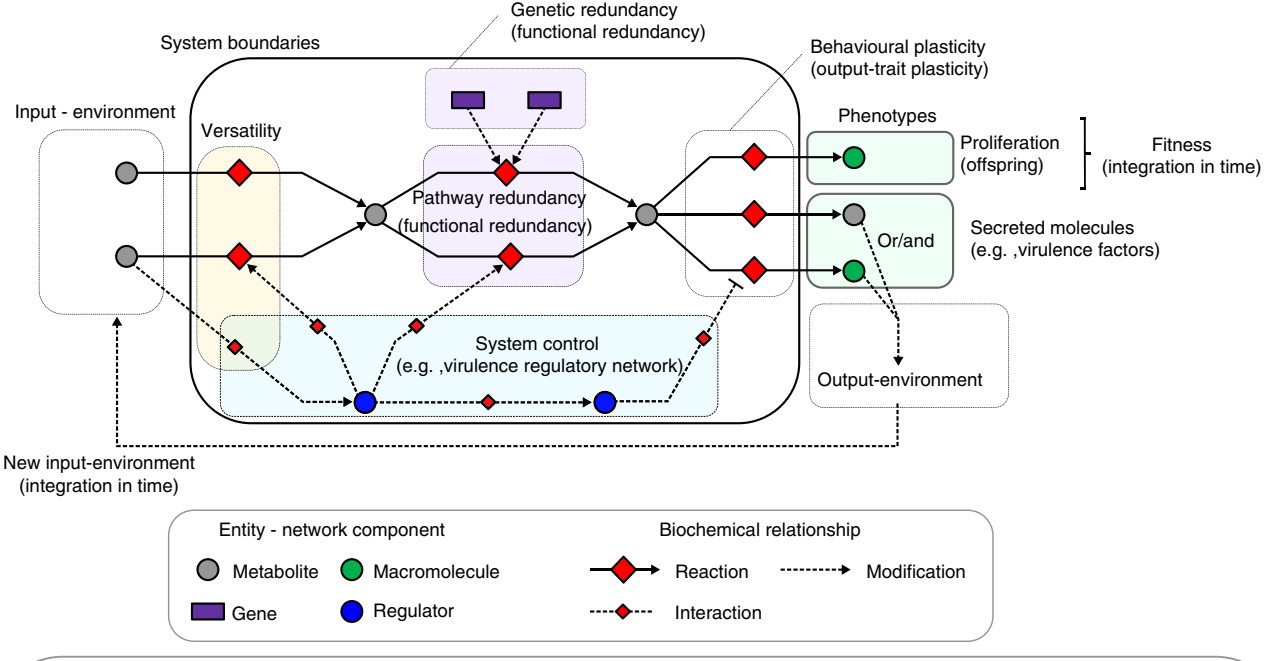

**Fig. 1** Sources of robustness in biological systems. The three main sources of phenotypic robustness in biological systems with respect to internal or environmental perturbations are: versatility, functional redundancy (genetic and pathway redundancy), and the system control. The virulence functions (virulence phenotypes) can be seen as a mechanism to modify the host environment to make it beneficial for the growth of the pathogen

metabolic transcriptional reprogramming can lead to lower fitness with regard to the kind of environmental perturbation encountered by an organism[28,29]. Interactions between the regulatory and metabolic networks therefore determine the properties of the global network and the distribution of robustness or fragility points.

One highly complex biological system in which robustness is a crucial property is the host-pathogen interaction[13]. During interaction, the pathogen subverts the host cell machinery to collect host resources[30], while the host reacts to protect its own physiological functions[31]. Each one will induce perturbations to the other's system, with the aim to lessen the system's robustness of the confronting organism[13,32]. Virulence functions of pathogens are controlled by sophisticated, multicomponent, interconnected regulatory networks that are responsive to environmental and internal signaling[33,34]. Reconstruction of metabolic models of pathogens and studies of their robustness with regard to internal perturbations were conducted with the aim to discover new drug targets[35,36] but only one virulence regulatory network has been reconstructed so far[37]. To our knowledge, robustness analysis at the system level of a pathogen combining both metabolic and virulence regulatory networks has not yet been investigated. This information should provide new insights about how phenotypic robustness is associated to the evolution of virulence determinants.

In this study, we investigated impact of the virulence regulatory network (VRN) on the phenotypic robustness of the plant pathogen *Ralstonia solanacearum*. This bacterium is one of the most destructive plant pathogenic bacterium worldwide due to its unusual wide host range and its broad geographical distribution[38–40]. Several reports have already highlighted that many pathogenicity functions display a high level of robustness as evidenced by genetic or functional redundancies[41–44]. We recently found that key VRN components can impair the versatility of the pathogen due to a resource allocation trade-off between virulence function and bacterial growth (hereafter referred as proliferation)[45]. Using a reconstructed cell-scale model of *R. solanacearum* and modeling studies, we predicted the robustness of phenotypes under environmental as well as internal perturbations. We evaluated the functional interplay between the metabolic network and the VRN and revealed the strong impact of this VRN on various sources of robustness such as functional redundancy. Results indicate that the VRN evolved to control functionally redundant metabolic genes, which overlap some primary metabolic pathways, thus leading to increased robustness capacities under specific host environment.

## Results

**Hybrid cell-scale model reconstruction of *R. solanacearum*.** We reconstructed a cell-scale model of *R. solanacearum* comprising three interconnected modules: a genome-scale metabolic network, a macromolecule network dedicated to virulence functions and a VRN (Fig. 2). The two first modules correspond to biochemical reaction networks previously described[45], whereas the VRN is a biochemical interaction network reconstructed in this study. Thus, the modular structure of this resulting "hybrid model"[46] allows computational approaches with appropriate methods for both types of networks, i.e., constraint-based modeling for the biochemical networks and multi-state logical modeling for the regulatory network, see Supplementary Note 1. Both methods do not require kinetic parameters and thus are relevant for genome-scale analysis.

*R. solanacearum* possesses a complex regulatory network which orchestrates the actuation of its large repertoire of virulence factors[40]. This VRN responds to various signals including host

cell contact, environmental nutrient availability, and quorum sensing molecules to coordinate the expression of multiple virulence factors at different steps of the infectious cycle. We reconstructed the VRN of *R. solanacearum* strain GMI1000 by collecting bibliographical information based on genetic and genomic studies (see Methods section for details), including three transcriptomic studies[47–49]. The reconstructed *R. solanacearum* VRN comprises 712 genes including 29 genes coding for transcription factors and 34 proteins involved in signal transduction. This VRN perceives 86 signals and controls the expression of 606 genes. Among those genes, 444 are present in the biochemical reaction network. Overall, 86 reactions of the macromolecule network are also regulated by the VRN, and correspond to the induction of macromolecule secretion via the control of secretion systems. In total, the VRN iRP1443REG encompasses 1443 interactions generating 705 logical rules (Fig. 3a, b, Supplementary Fig. 1, Supplementary Data 1). The resulting hybrid model includes a substantial proportion of the genes annotated in the reference genome as enzymes (71.1%), transporters (65.3%), regulators (34.3%), receptors (75%), and virulence genes (90.2%), indicating a significant level of completeness (Supplementary Table 1).

To validate the capacity of the reconstructed VRN to predict correctly gene expression in response to environmental changes, we compared the differential gene expression from an experimental transcriptomic data-set with the model predictions. These experimental data were extracted from a study using *R. solanacearum* grown either in complete medium in vitro or in planta during xylem colonization[50]. This experimental data-set was not used for the VRN reconstruction and was therefore well-suited to test the prediction capacity of the VRN module. The prediction of gene expression was performed through a discrete logical modeling method using FlexFlux[51] under various environmental conditions, including in vitro and in planta conditions (see Supplementary Data 2; Supplementary Note 2). Experimental gene expression values were discretized using a cut-off threshold of 2.0 of absolute log2 expression levels to be directly compared with the model predictions (Supplementary Data 3). Among 64 conditions tested, the differential gene expression predicted in xylem tissues at high-cell density (thus similar to the experimental condition) was the closest match with experimental gene expression data, F1 score 0.60 (Supplementary Data 3), see Fig. 3c and Supplementary Note 3. To assess the predictability of the network and the sensitivity of each environmental component to trigger the specific gene transcription reprogramming, we evaluated the deviation of the prediction induced by changing each environmental component (see Methods). Simulations revealed four key environmental constraints acting on the VRN status listed in the order of importance: (i) the plant cell wall sensing, (ii) quorum sensing (i.e., high-cell density versus low-cell density), (iii) $O_2$ limitation, and (iv) nitrate availability. We concluded from this analysis that the model reliably predicted the transcriptional responses controlled by the VRN and the major signals or constraints it perceives in its environment.

We tested the capacity of the hybrid model to predict the modulation of the phenotypic plasticity by the VRN. We performed a simulation of various phenotypes in different environmental conditions and compared them with experimental observations (list in Supplementary Data 4). In addition, we compared the predicted and experimentally observed phenotypes of 16 VRN deletion mutants (Supplementary Data 4). We found an overall accuracy of 77% of the cell-scale model prediction for phenotypic plasticity (Fig. 3d). For instance, simulations reproduced the well-known phenotypic switch induced by the quorum sensing system[52,53], which controls many biological

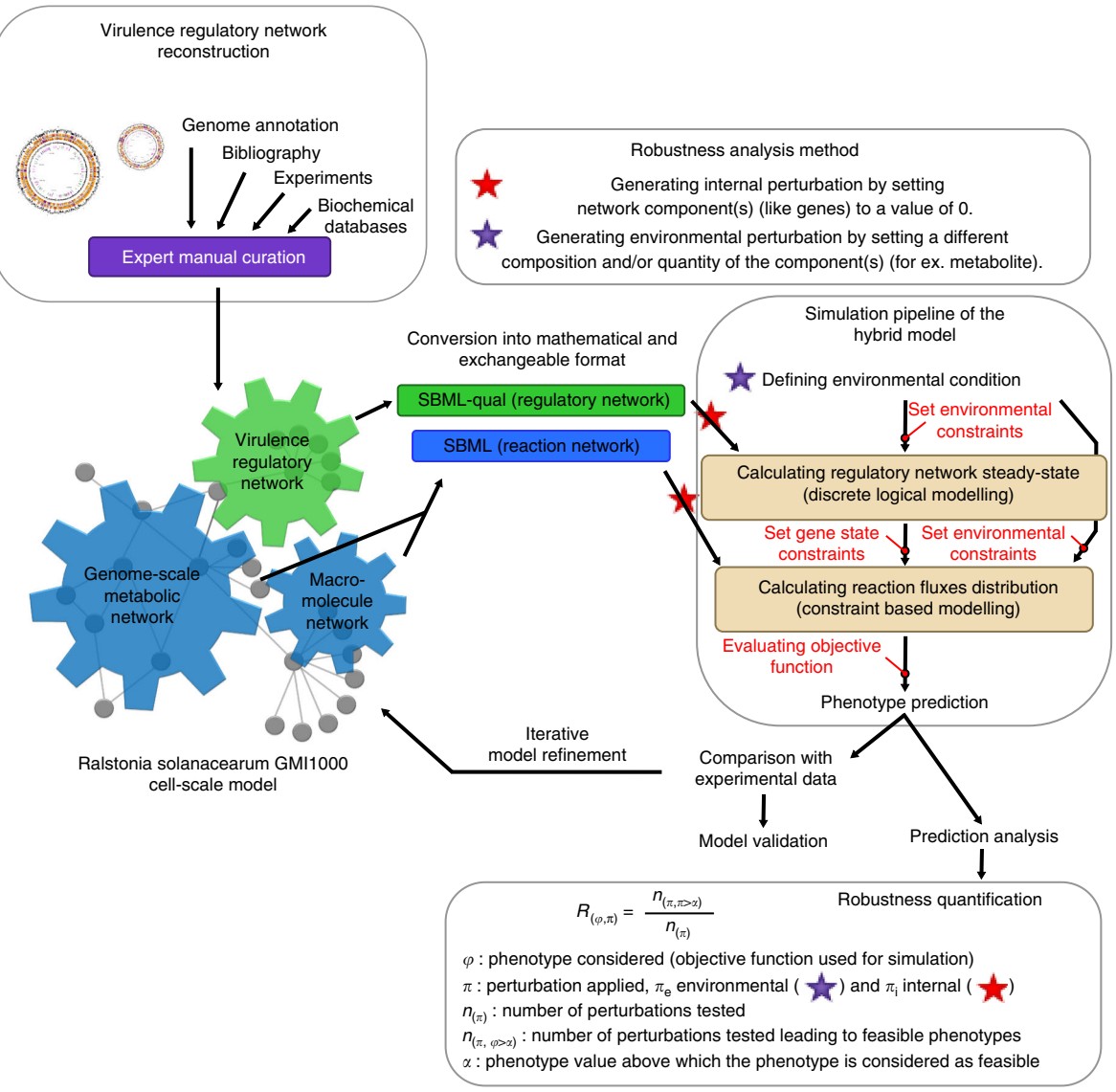

**Fig. 2** Architecture of the cell-scale model of *Ralstonia solanacearum* GMI1000 and the simulation pipeline for robustness analysis. The cell-scale model is composed of three modules: a genome-scale metabolic network, a macromolecule biosynthesis and secretion module[45], and the virulence regulatory network (VRN). Simulation was performed using hybrid modeling, i.e., discrete logical modeling for the regulatory network and constraint-based modeling for the biochemical reaction network. Robustness quantification was performed using the BECO analysis, which evaluates impact of environmental perturbations (πe) or internal perturbations (πi) on phenotypes

functions including production of virulence determinants or motility repression.

The capacity of the metabolic model to predict the functional redundancy of the pathogen metabolism was experimentally assessed by screening a Tn5 transposon insertion mutant library. A bank of 4046 random transposon insertions in *R. solanacearum* strain GMI1000 was generated and screened for gene essentiality under two growth conditions, i.e., D-glucose and L-glutamate as sole carbon and energy sources (see Methods, Supplementary Data 5, and Supplementary Note 4). Among the 176 genes present in the metabolic model and in the mutant library, 146 of them harbored a deletion phenotype accurately predicted by the model in both conditions, with an overall accuracy of 83% (see Supplementary Data 6). Among the predictions, which were confirmed experimentally, is the functional redundancy between the two copies of the malic enzymes and the phosphoenolpyruvate carboxykinase, which catalyze reactions required during growth on L-glutamate (see Supplementary Data 6).

**Quantification and sources of phenotypic robustness**. The challenge in apprehending robustness properties of biological networks comes from the difficulties in defining a quantitative metric of robustness[1,3]. We set out a metric that could be broadly used and results in the probability of the phenotype to be maintained in presence of defined perturbations. We defined the robustness ($R_{(\varphi,\pi)}$) of the phenotype ($\varphi$) facing the set of perturbations ($\pi$) as the number ($n_{(\pi,\varphi>\alpha)}$) of perturbations for which the phenotype is feasible, i.e., the value of the phenotype is strictly superior to a chosen threshold ($\alpha$), divided by the number ($n_{(\pi)}$) of the faced perturbations.

$$R_{(\varphi,\pi)} = \frac{n_{(\pi,\varphi>\alpha)}}{n_{(\pi)}}. \qquad (1)$$

Hence, this robustness value ($R$) ranges from 0, when each perturbation leads to a loss of function, to 1 when the phenotype is always maintained in presence of any perturbation tested. This

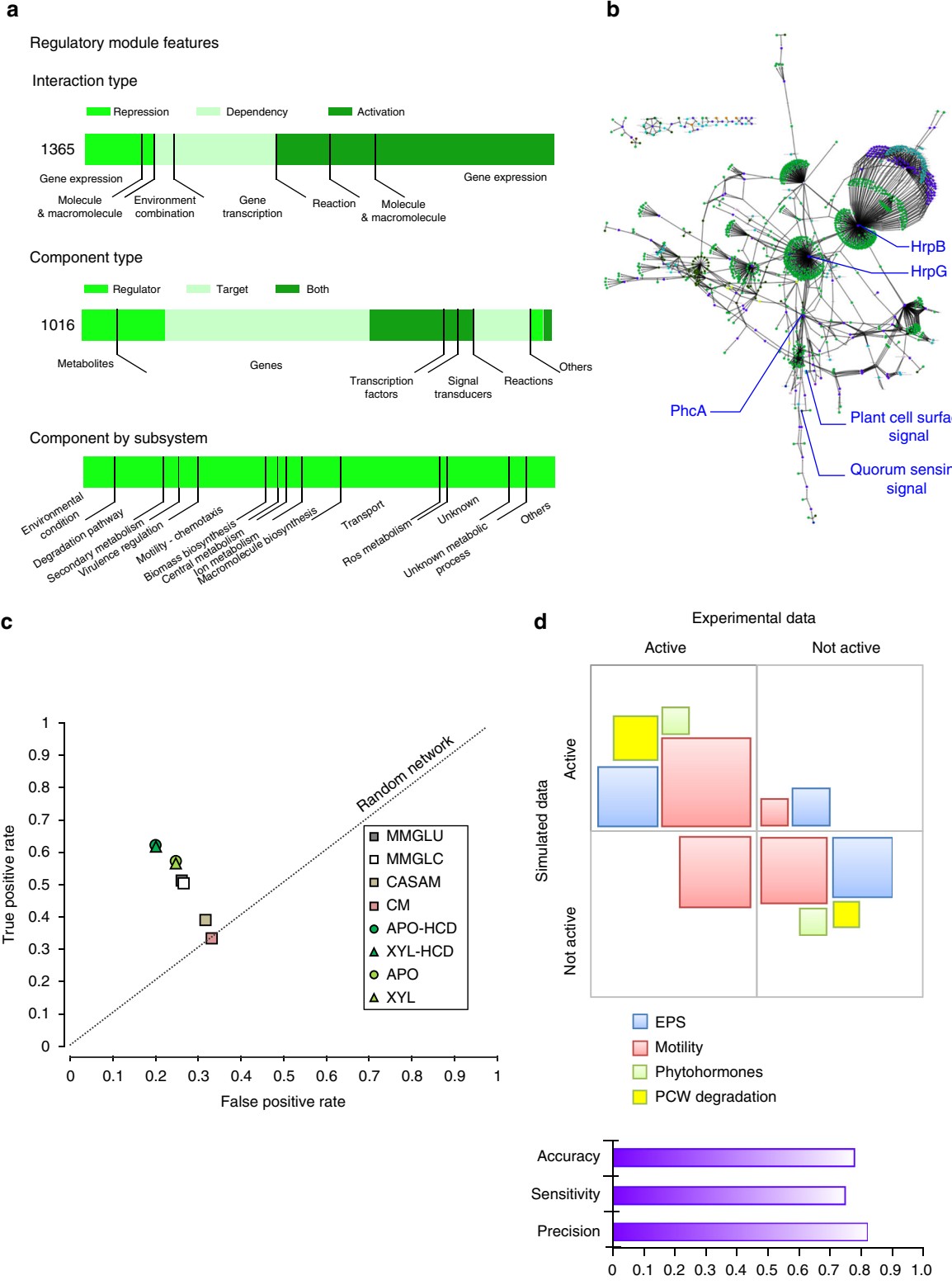

**Fig. 3** Properties of the VRN and prediction performance. **a** Compositions and metrics of the virulence regulatory network. **b** View of the VRN structure with the three major regulatory hubs: PhcA, HrpG, HrpB (see detail in text). **c** Prediction capacity of the VRN on differential gene expression compared by an accuracy analysis to an in planta transcriptomic data set[50]. Environmental conditions used for simulation: MMGLU minimal medium with ʟ-glutamate, MMGLC minimal medium with ᴅ-glucose, CASAM minimal medium with casamino acids, CM complete medium, APO tomato apoplast, XYL tomato xylem, HCD high-cell density. **d** Phenotype prediction of the cell-scale model (biochemical reaction network plus regulatory network) compared to experimental data for four groups of phenotypes and VRN deletion mutants in various conditions. The term "active phenotype" means that the corresponding biological function was either observed experimentally or inferred from simulation of the cell-scale model (i.e. possibility to realize a metabolic flux through the reaction associated to the phenotype). Size of the boxes is proportional to the number of phenotypes in each category ($n = 43$). The metrics of the model performance are reported as precision, sensitivity and accuracy. EPS Exopolysaccharide production, PCW plant cell wall

definition is prone to each considered phenotype and perturbation. It has also the advantage to allow comparison of robustness of different phenotypes with respect to a similar set of perturbations as well as comparing the phenotypic robustness of different organisms encountering a similar set of perturbations. To perform in silico robustness quantification, we designed a global robustness analysis, called BECO for Biological Entity Clustering by their contribution to the Objective function. This algorithm performs multiple regulatory steady-state analyses[51] followed by FBA of a set of phenotypes ($\phi$), used as objective functions, under a number $n_{(\pi e)}$ of environmental conditions ($\pi e$) (see Methods and Supplementary Note 5 for details).

In this study, the BECO analysis that was conducted referred to the following definitions: a phenotype is considered to be feasible if a flux through its specific reaction or defined set of reactions ($\phi$), i.e., an objective function, is possible ($\phi > \alpha$, with $\alpha = 0$ in our study). An environment refers to different compositions and concentrations of compounds available in the medium and to physical parameters (e.g., temperature). An environmental perturbation is simulated by changing composition or concentration of any compound in an environment or any physical parameter. An internal perturbation is simulated by removing a metabolic gene, a regulator or a reaction.

Then, BECO systematically tests the contribution of each biological entity (genes, regulators, and reactions), i.e., internal perturbations, on the phenotypes under several environments that correspond to environmental perturbations. This is realized by simulating series of gene and reaction knockout (KO) analyses and flux variability analyses, which leads to a classification of these entities similarly to the classification proposed by Wang and Zhang[10] and Lewis et al.[27] (see Methods).

**Impact of the VRN on robustness upon external perturbations.** As a first analysis, we assessed if *R. solanacearum* has a level of phenotypic robustness comparable to what is observed for other microorganisms. Hence, we determined the level of genetic redundancy in *R. solanacearum* and 10 other representative microorganisms. We also performed a BECO analysis on orthologs shared between *R. solanacearum*, *E. coli*, and *P. aeruginosa*. We found that both levels of functional and genetic redundancy

ensuring robustness of the proliferation trait were similar in *R. solanacearum* and the other tested bacteria, see Supplementary Figs. 2 and 3, and Supplementary Note 6.

We then investigated the impact of the *R. solanacearum* VRN on phenotypic robustness when bacteria face an environmental perturbation. We selected a range of 14 phenotypes (listed in Fig. 4a) representative of a broad set of functions performed by bacteria, including housekeeping functions (proliferation, production of storage compounds) and known virulence and/or plant-associated functions (Type 3 effector secretion, exopolysaccharide production, phytohormone production, motility and secretion of plant cell wall degrading enzymes). The set of 14 environmental perturbations ($\pi e$) tested were representative of several environmental conditions (i.e., different carbon and nitrogen sources or presence of quorum sensing signals, etc…) encountered by the bacteria in vitro or in planta (see Supplementary Data 7). Robustness$_{(\rho, \pi e)}$ of the phenotypes calculated by the BECO analysis, with or without considering the VRN, is presented in Fig. 4a. These simulations revealed that the different phenotypes tested followed distinct patterns: the virulence-associated functions were subjected to a strong reduction of their robustness with respect to environmental changes due to the VRN (up to 60% of reduction of their solution space) whereas housekeeping function phenotypes in similar conditions retained robustness. This difference was not due to a difference between the number of VRN-regulated genes involved in the housekeeping ($14.3 \pm 2.6\%$) and virulence phenotypes ($15.1 \pm 4.5\%$), $p$-value 0.628. It cannot be ruled out however that some redundant genes involved in housekeeping functions may be also controlled by regulatory networks other than the VRN, and thus could become essential under specific environments.

To assess the significance of the observations made through the BECO analysis, we developed a 2D robustness analysis (ROBA), which simulates robustness of the phenotypes when considering a broader range of 1000 random environmental perturbations (see Supplementary Note 7). Here again, impact of the VRN on phenotypic robustness had a strong effect on virulence-associated functions but not on housekeeping functions, see Fig. 4b. This confirmed that in presence of the VRN the virulence-associated and housekeeping functions can be differentiated by their

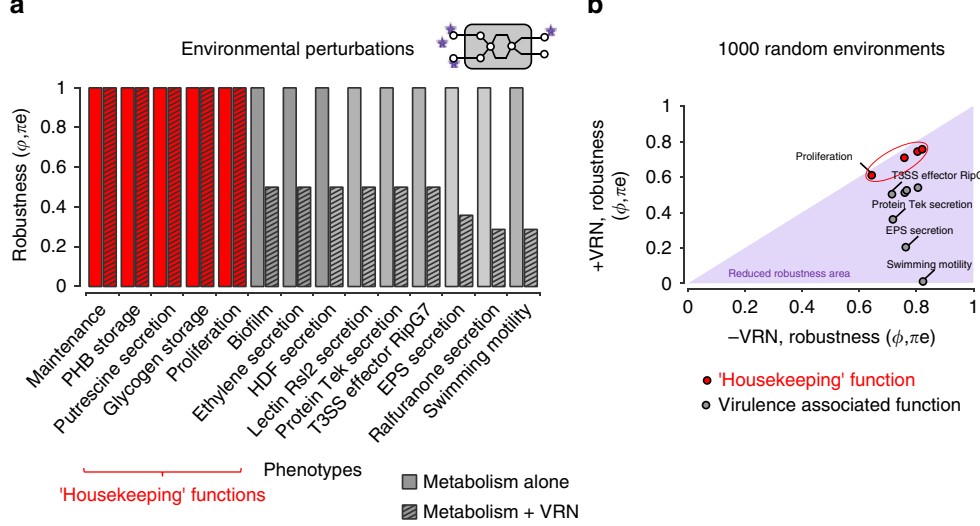

**Fig. 4** The VRN restricts phenotypic robustness in presence of environmental perturbations. **a** Impact of the VRN on the phenotypic robustness in presence of environmental perturbations. The simulations were conducted in presence or absence of the VRN in 16 different environments, including in planta (e.g., plant tissues, high and low-cell density) and in vitro (e.g., complete medium, minimal medium) conditions. The detailed list of tested environments is provided in Supplementary Data 7. **b** Analysis of the impact of the VRN on phenotype robustness after simulation in presence of 1000 random environments

sensitivity to environmental perturbations ($t$-test, $p$-value $6 \times 10^{-4}$). The apparent decrease of robustness for the virulence-associated phenotypes indicated that essential genes required for virulence were not induced or repressed by the VRN in some specific environments. When comparing conditions that activate virulence functions, we found that the robustness of the virulence phenotypes is higher in planta ($0.741 \pm 0.313$) compared to in vitro conditions ($0.208 \pm 0.140$), $p$-value 0.0006. On the other side, housekeeping functions were almost insensitive to the VRN control with respect to environmental perturbations, thus showing that expression of essential genes required for housekeeping functions is not repressed by the VRN in any of the tested environments.

**The VRN plugs on redundant genes of the primary metabolism**. The previous analysis revealed that phenotypic robustness of the housekeeping functions in presence of environmental perturbations is insensitive to the VRN control. However, it appears that among the 444 metabolic genes under control of the VRN in the reconstructed model, a large number of them code for primary metabolism enzymes, i.e., involved in the biosynthesis of cell biomass[45,46]. This observation suggested that the VRN-regulated genes coding for primary metabolism components can be sources of robustness through genetic redundancy or pathway redundancy. We used the BECO method to test the contribution of these VRN-regulated genes in the primary metabolism to the robustness of the proliferation phenotype. Hence, the robustness of various biological functions was assessed with respect to internal perturbations ($\pi i$), i.e., when occurs a loss of function of each gene or reaction included in the model, in a defined set of 14 different environments (see Supplementary Data 7). This simulation mimicked the occurrence of a deleterious mutation or the inhibition of an enzymatic reaction. We found that proliferation is the function the most impacted by the VRN when internal perturbations occur, thus indicating that a significant number of metabolic genes providing robustness to this phenotypic trait are under the control of the VRN (Supplementary Fig. 4, and Supplementary Note 8 for detailed analysis).

We verified that the connection of the VRN on metabolic pathways inferred from the BECO analysis was significant and therefore statistically different from a random connection. We first defined eight different classes of genes, based on how they contribute to robustness of the metabolic network (Fig. 5a), see Methods for the procedure used for gene class definition. These different classes also distinguish the contribution to robustness that are conditionally dependent (i.e., dependent upon a specific environmental condition). The output of the BECO analysis then provided for each class the proportion of genes in the network, with simulations either in presence or absence of the VRN (Fig. 5b; Supplementary Data 8). We tested if the gene distribution in the different classes was dependent on VRN control by performing $\chi^2$ tests. The statistical analysis showed that the connection of the VRN into the metabolic network is not random ($p$-value of $1.9 \times 10^{-6}$ and $4.1 \times 10^{-4}$ for the proliferation and T3SS effector production phenotypes, respectively). A tendency to an enrichment of VRN-regulated genes in conditionally dependent redundant pathways was observed (see classes OPT-C and ELE-C in Fig. 5b). Other virulence-associated traits followed the same trend, such as secretion of lectin Rsl2, extracellular proteins Tek and Pme with a $p$-value of $4.1 \times 10^{-3}$, $6.7 \times 10^{-4}$ and $3.5 \times 10^{-3}$, respectively; or EPS production with a $p$-value of $1.9 \times 10^{-2}$. Interestingly, the pattern of gene distribution in the different classes for the VNR-regulated genes differed between phenotypes, suggesting that different metabolic constraints occur depending on the phenotype. For instance, the connection of the VRN within the primary metabolism to control

proliferation (housekeeping function) differs from T3SS effector production (virulence-associated function) ($p$-value $5.2 \times 10^{-5}$), see Fig. 5.

Next, we ran a BECO analysis to identify under which conditions were mobilized the VRN-regulated genes that provide $\pi i$-robustness for the two phenotypes (T3SS and proliferation). We determined the Robustness under three conditions: an environment not activating the VRN used as control, and two environments with activating signals encountered by the bacteria during infection: the presence of plant cell wall components[54] and the quorum sensing signal[53]. The simulations predicted that the robustness of both phenotypes increased when the VRN was activated, either by the plant cell wall or the quorum sensing signal (Supplementary Fig. 5). The decrease of the number of essential genes indicated that other network components were activated: the activation of redundant genes or pathways by the VRN could complement some reactions which are otherwise essential when the VRN is not activated. This analysis also suggested that phenotypic robustness is higher in planta compared to in vitro conditions. We conducted simulations to test the robustness of the proliferation function in different environments: this indeed revealed that robustness was significantly higher in planta ($0.783 \pm 0.004$, for six environments tested) compared to in vitro ($0.751 \pm 0.016$, for eight environments tested), $t$-test $p$-value 0.0005.

**VRN-regulated genes acquired via horizontal gene transfer**. Our analysis revealed that several VRN-regulated genes within the primary metabolism provide robustness to proliferation as well as virulence functions. To get evolutionary insights on how phenotypic robustness was shaped by the VRN, we performed phylogenetic analyses using homology groups containing the genes from 28 representatives of the beta-Proteobacteria (including *R. solanacearum*) and some gamma and alpha-class species as outgroups (Supplementary Table 2). We focused our analysis on the amino-acid biosynthesis pathways, which are primary metabolism pathways broadly shared between organisms. Many of these pathways possess VRN-regulated genes and contribute to phenotypic robustness for both the proliferation and virulence functions. We compared the presence of homologs of the *R. solanacearum* genes providing robustness in those organisms, as well as the presence of homologs of key regulators of the VRN, like *hrpG* and *hrpB*[40]. We found that the VRN plugs into six different *R. solanacearum* amino-acid biosynthetic pathways and mainly controls genes likely acquired via horizontal gene transfer (Supplementary Fig. 6). This is illustrated with the tryptophan biosynthesis pathway: two reactions of the pathway are catalyzed by redundant enzymes, some of them being regulated by the VRN (Fig. 6a). The distribution of the genes encoding these enzymes was performed among the 28 Proteobacteria and this analysis revealed that only the *R. solanacearum* species possesses the VRN-regulated genes among beta-Proteobacteria. The VRN-independent gene *trpD1* (locus tag RSc2884) belongs to the highly conserved tryptophan biosynthesis operon found in most bacterial species and is likely the most ancestral gene (Fig. 6b). The VRN-regulated gene *trpD2* (locus tag RSp0681) is suspected to have been acquired through horizontal gene transfer as it is not found in others beta-Proteobacteria outside of the *R. solanacearum* species and a blast analysis first identified an homologous gene in alpha-Proteobacteria (62% identity at the protein level with *trpD2*) instead of *R. solanacearum trpD1* (45% identity). This observation was supported by a phylogenetic analysis of *trpD* sequences, which revealed incongruence on taxonomy for *trpD2* (Fig. 6c). For another reaction of the pathway, the VRN-regulated gene *trpC2* (RSp0680) also reveals incongruence of the phylogeny with the taxonomy (Fig. 6b;

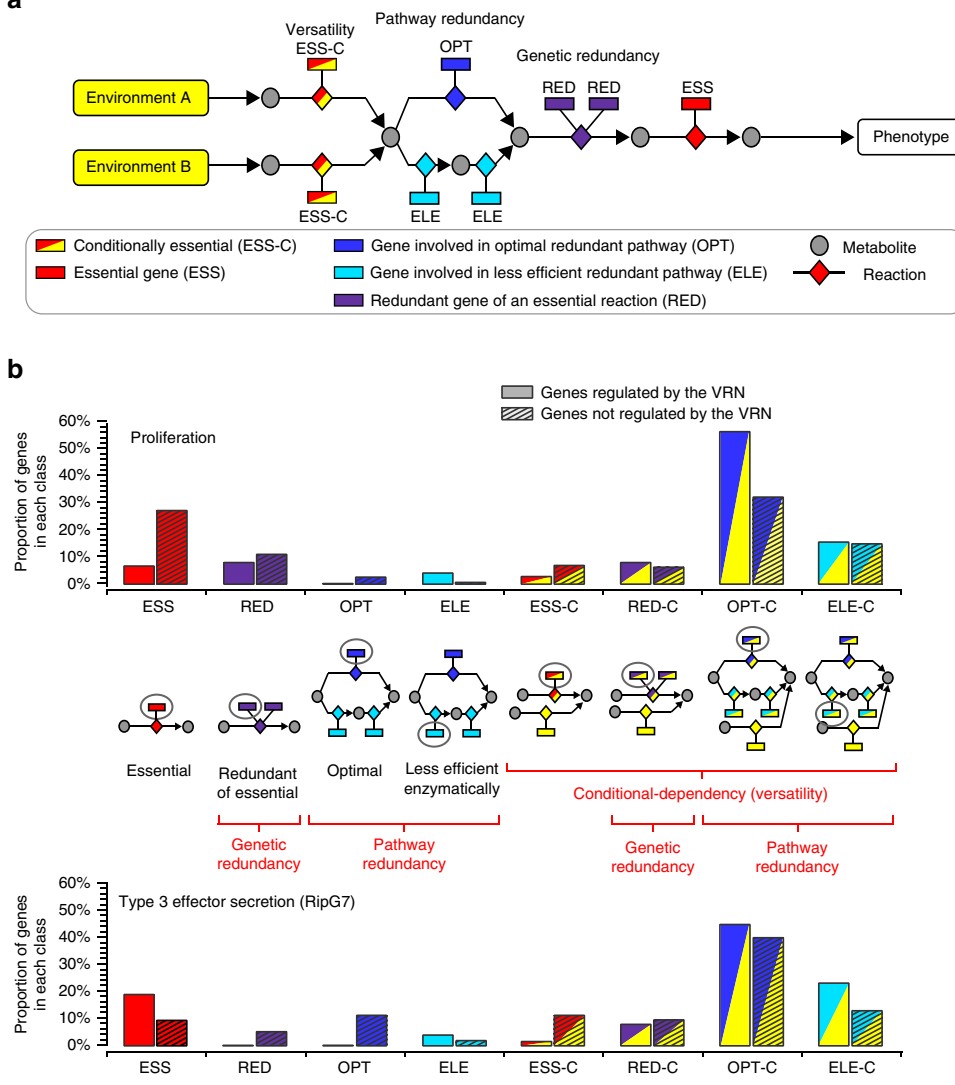

**Fig. 5** Sources of phenotypic robustness mobilized by the VRN with respect to internal and external perturbation. **a** Sources of phenotype robustness in a metabolic network. **b** Classification and distribution of genes contributing to robustness of the proliferation phenotype ($n = 78$ for VRN-regulated genes and 839 for VRN-independent genes) and the T3SS-dependent secretion of the effector RipG7 ($n = 78$ for VRN-regulated genes and 521 for VRN-independent genes) obtained with the BECO analysis. Dependency of the two distributions, VRN-regulated vs VRN-independent, is shown and significance of the differences was assessed by performing a $\chi^2$-test. In total 919 metabolic genes were classified by the BECO analysis

Supplementary Fig. 7) and best blast search pinpoints homologs in alpha-proteobacteria (*Methylocystis*; 55% identity) and gamma-Proteobacterium (*Pseudomonas syringae*; 54% identity) rather than the *R. solanacearum* ancestral gene *trpC1* (RSc2885, 48.4% identity). These data were again suggestive of a horizontal acquisition of *trpC2*. A similar situation was also observed for other VRN-regulated genes in the L-proline (RSp0418), L-Lysine (RSp0424), L-cysteine (RSp0417), L-aspartate (RSp0943), and L-methionine (RSp0781) biosynthesis pathways, see Supplementary Fig. 6. All these VRN-regulated genes provide genetic redundancy, except RSp0781 in the L-methionine biosynthesis pathway, which illustrates a case of pathway redundancy.

Phylogenetic relationships analyses of the VRN-regulated genes in amino-acids biosynthesis pathways revealed an absence of orthologs in other beta-Proteobacteria (Supplementary Fig. 6). The hypothesis of a horizontal gene transfer origin for the acquisition of the VRN-regulated genes is also supported by a clear enrichment of VRN-regulated genes present only in *R. solanacearum* strains among beta-Proteobacteria: homology

searches revealed that 30% of the VRN-regulated genes are specific to the *R. solanacearum* species whereas the proportion is only 3% for the VRN-independent genes (Supplementary Fig. 8). Moreover, a bias in GC content was detected in the total number of VRN-regulated genes compared to the VRN-independent genes, *p*-value $3 \times 10^{-4}$ (Supplementary Fig. 9; Supplementary Data 9), again suggestive of horizontal acquisition. Several other VRN-regulated genes involved in primary metabolism pathways, such as the TCA cycle, were also identified to provide robustness patterns, and also presumably acquired through horizontal transfer, see Supplementary Data 10.

## Discussion

Phenotypic robustness is a remarkable property of biological systems, allowing them to maintain their functions with respect to perturbations[1,3,55]. Here we developed a system modeling-based approach to study how phenotypic robustness upon internal and environmental perturbations is reprogrammed by

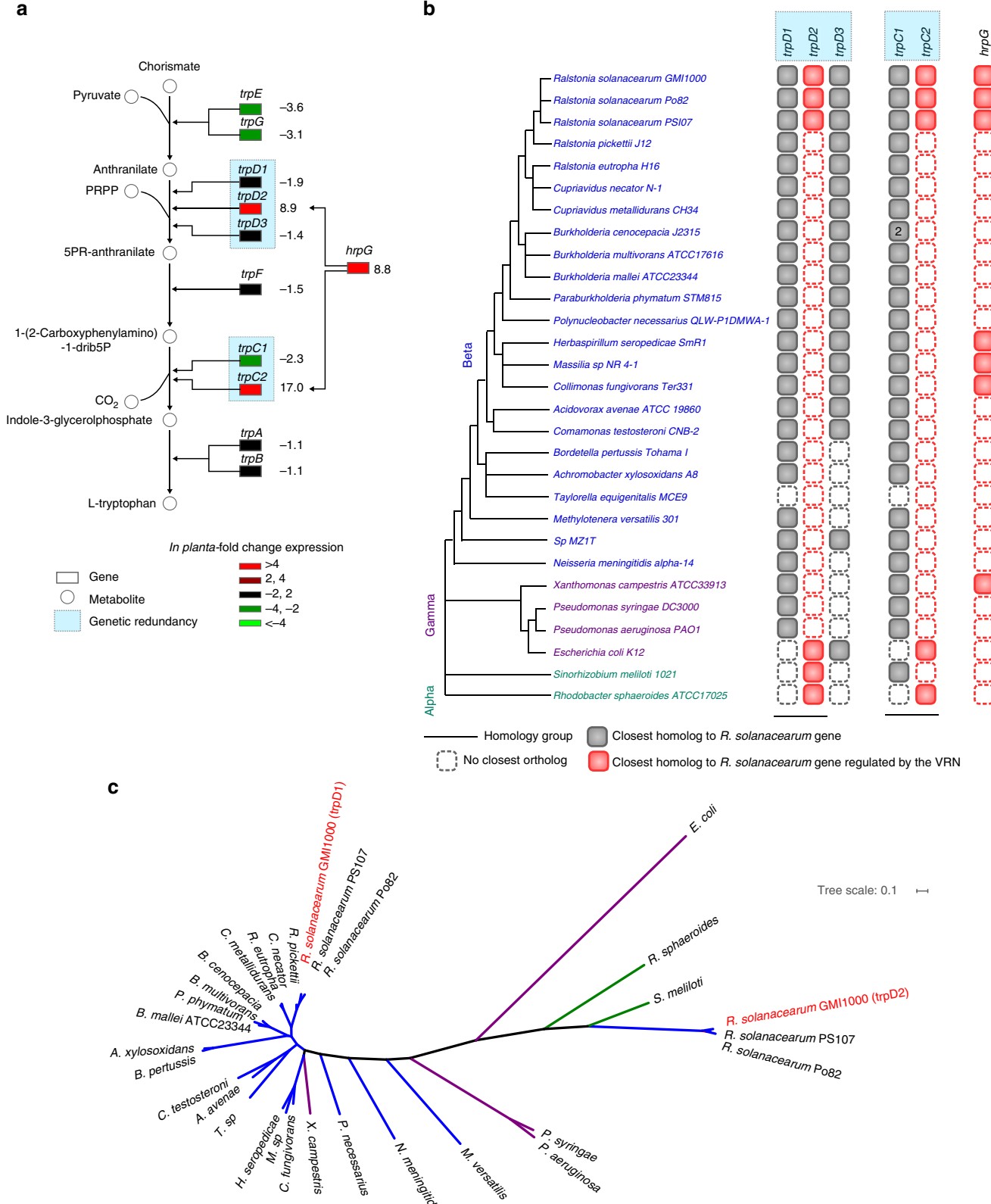

**Fig. 6** Orthology analysis of VRN-regulated and VRN-independent genes contributing to robustness within the tryptophan biosynthesis pathways. **a** The *R. solanacearum* tryptophan biosynthesis pathway and the corresponding in planta gene expression profiles. The values correspond to the absolute fold change expression levels (from Jacobs et al. [50]). **b** Presence of homologs of the *R. solanacearum* genes in the various taxa is displayed by boxes. Number in the boxes corresponds to the number of paralogs present in the organism. Closest homologs to a VRN-regulated gene are colored in red, whereas the closest homologs of a VRN-independent one are in gray. In the case of two genes belonging to the same homology group, the closest homolog to a *R. solanacearum* gene was defined as the gene showing the highest blast score to a *R. solanacearum* gene. **c** Phylogeny analysis of the *trpD1* and *trpD2* homolog groups. Tree scale corresponds to 0.1 mutation

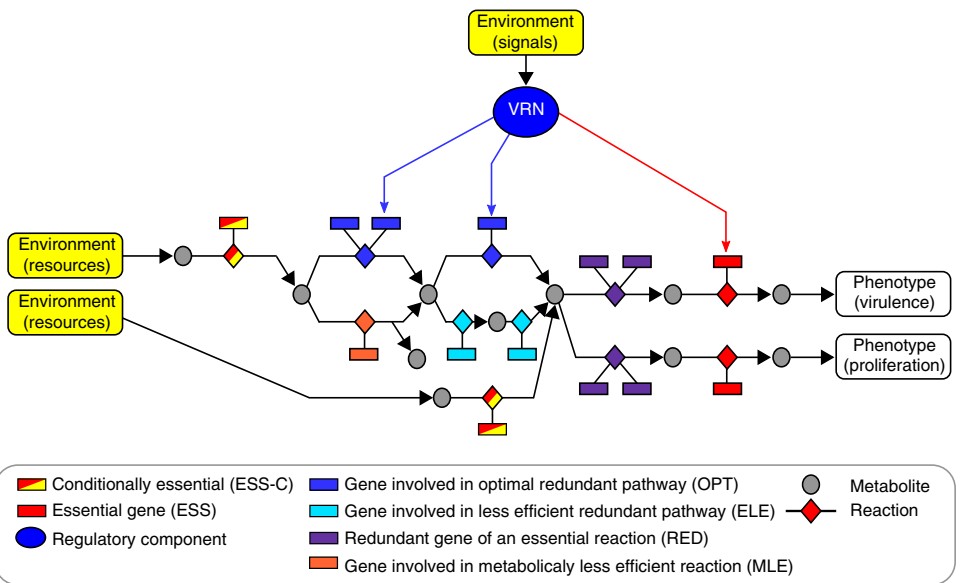

**Fig. 7** Functional connections of the VRN with the metabolic network and their impact on the control of phenotypic robustness. Rectangles, diamonds, and circles symbolize genes, enzymatic reactions, and metabolite products, respectively. The action of the VRN is either on the control of robust nodes of the primary metabolic network (blue arrows) or on essential required for expression of virulence phenotypes (red arrow). Examples of the first class (genes colored in blue) are the tryptophan biosynthetic pathway (genetic redundancy) and the methionine biosynthetic pathway (pathway redundancy), see details in text

the VRN in the plant pathogen *R. solanacearum.* The proliferation in the literature of the term "robustness" applied to biological systems has led to inappropriate or confusing usage of this term[3,56]. As stated before, the notion of robustness requires to specify which phenotypic trait is robust, to which perturbation, and to provide quantification. We provided a metric useful for a quantitative estimation of the robustness of phenotypes with respect to perturbations. Such a metric was lacking to perform deep comparisons of phenotypic robustness within biological systems[1]. Our metric has the advantage to be generic in the sense that it can be applied to any phenotype of any biological system facing any perturbation. The use of this robustness metric was associated with the development of a bioinformatics pipeline to calculate the impact of the internal and/or the environmental perturbations on several biological functions at genome-scale level.

A second achievement of this study was the reconstruction of a *R. solanacearum* genome-scale cell model integrating a metabolic network, a macromolecule secretion module and a regulatory network. An originality of this hybrid cell model is the implementation of a VRN comprising 712 genes, which is responsive to 86 environmental signals. Thus, this is one of the most complex VRNs reconstructed so far in bacterial pathogens[40], comparable in complexity to the *P. aeruginosa* VRN[33,37]. The quality and performance of this hybrid cell model was validated through different assessment steps, which include gene essentiality analyses using a transposon insertion mutant library, comparison with -omics data sets and mutant phenotyping.

The large repertoire of genes regulated by the *R. solanacearum* VRN (up to 444 into the metabolic and macromolecule modules) opened an unprecedented possibility to evaluate how metabolic network components from a pathogen are specifically controlled by the VRN. Our results reveal that the *R. solanacearum* VRN controls phenotypic robustness through two different means: (i) as expected, by activating essential reactions required to produce virulence phenotypes in specific environment, (ii) more surprisingly, by activating functionally redundant reactions within

primary metabolism, presumably with the aim to specifically sustain virulence (see Fig. 7). Obviously in the first case, the VRN optimizes the network state to avoid the expression of costly virulence functions (such as the EPS biosynthetic pathway) in inappropriate conditions. Such a resource allocation trade-off between virulence and proliferation exists in *R. solanacearum,* due to a probable limitation of nutritional resources in the host environment[45]. In addition, the biosynthesis of biomass and virulence-associated traits involves many common metabolic precursors. The burden for the primary metabolism in supporting simultaneously multiple objectives therefore imposes an optimization constraint on the activation of virulence pathways, a process tightly regulated by the VRN[26,45,50]. This explains why control exerted by the VRN could restrict in a large extent the phenotypic robustness of virulence functions with respect to environmental perturbations. Because the *R. solanacearum* global regulatory network (including metabolic regulation, stress response, etc…) is not yet reconstructed, the impact of this network could not be evaluated. However, this study revealed the direct effect of the VRN on phenotypic robustness as the analyses were conducted in a straight comparison between activated/ inactivated VRN states independently of the global regulatory network; future investigations are required to undercover potential cross talks between the *R. solanacearum* VRN and the global regulatory network.

The resource allocation trade-off between virulence and proliferation traits may also generate perturbations on the regulation of some primary metabolic pathways as the corresponding genes are subjected to the basal metabolic regulation dedicated to the homeostasis of biomass precursors, but also to regulation by the VRN. Interestingly, the VRN plugs on different genes that apparently overlap genes or operons involved in amino-acids biosynthesis. This control therefore provides phenotypic robustness to internal perturbations during infection and could also be a potent way to increase some metabolic fluxes required for the efficient synthesis of virulence factors. There is indeed increasing evidence for the contribution of primary metabolism genes to

support both virulence and growth traits in various bacterial pathogens[57–60], and our report suggests that some primary metabolic pathways are specifically relevant for virulence purposes in *R. solanacearum*.

Phylogenetic analyses support the view that many of these primary metabolism VRN-regulated genes were probably acquired via horizontal gene transfer to supply a functional overlay to ancestral genes. This appears to be the case for several amino-acid biosynthetic pathways (tryptophan, proline, lysine, cysteine, and aspartate). Such apparent redundancy allows uncoupling the VRN control from the basal homeostatic control on these amino-acid biosynthetic pathways. This probably provides an advantage for the pathogen and may compensate the cost of possessing two functionally redundant genes. This process of metabolic gene acquisition and recruitment in the virulence regulon thus appears to be a key mechanism contributing to the phenotypic robustness of *R. solanacearum*. Another situation is observed in the case of the methionine biosynthetic pathway: here the *metE* gene encoding the cobalamin-independent methionine synthase was recruited by the VRN, whereas the functionally redundant *metH* gene encoding the cobalamin-dependent enzyme was insensitive to VRN regulation (Supplementary Fig. 6). MetE provides robustness to proliferation because it was previously shown to be required for full pathogenesis, being critical for infection as cobalamin is not produced by the plant host[61]. Significantly, our analysis shows that the VRN-regulated genes in amino-acid biosynthetic pathways, which provide robustness were most probably acquired through horizontal acquisition rather than duplication of ancestral genes. This highlights horizontal gene transfer as a major mechanism promoting evolution of virulence and suggests that robustness is shaped through evolutionary selection.

The robustness of biological systems implies that a limited number of internal perturbations can substantially disrupt the system[13] and the identification of sources of robustness of pathogens will be a major issue for medical or agricultural applications. The *R. solanacearum* cell model is a valuable hypothesis-testing tool for further functional approaches in a biological system where phenotypic robustness potentially hampers the identification of virulence/plant adaptation determinants[32].

## Methods

**Construction of a Tn5 transposon insertion mutant library**. A *R. solanacearum* mutant database was generated through random insertion of the transposon EZ-Tn5<KAN-2> (EPICENTER® Biotechnologies) in strain GMI1000[39]. The EZ-Tn5<KAN-2> was electroporated in strain GMI1000 competent cells following the protocol of the supplier and mutants were isolated on BG medium supplemented with Kanamycin. Localization of each individual insertion in the genome was determined by sequencing of its flanking region (which may differ from the mutated gene) using an adaptation of the TAIL (Thermal asymmetric interlaced)-PCR protocol from Qian et al.[62] and automated BlastN comparison with the genome sequence. A total of 4046 transposon insertions in the GMI1000 genome were accurately mapped (Supplementary Data 5) and their list and position can be visualized on a dedicated webpage: https://iant.toulouse.inra.fr//bacteria/annotation/site/prj/ralso/tools/mutants_db/cgi/EZLucene.cgi?lucenedb=mutants&simpleform=1.

**Virulence regulatory network reconstruction**. The VRN was reconstructed following the recommendations for Transcriptional regulatory network reconstruction[63] but extended to consider the entire set of components involved in a regulatory process, i.e., from signal molecule to transduction cascade and transcription activation. The reconstruction was performed using bibliographic information based on genetic and genomic studies, including three transcriptomic studies available in the literature[47–49]. This model includes the list of interactions (activation or repression) involving one regulator and one target such as the binding of transcription factor to target gene promoters, the activation of a receptor (protein) by metabolites, protein-protein interactions, etc…. These various kinds of interactions were annotated following Systems Biology Ontology recommendations[64]. The interactions also contain information of the state of the regulator

leading to the change of the target status. Indeed, when a quantitative value is available, like affinity of a receptor for a metabolite or quantitative response of the target to a transcription factor, the value is converted into discrete thresholds (discrete logical relationship). The interactions were then converted in logical rules describing for each target the rule of activation, i.e., state of the regulators and their combination leading to the change of the target state. Then, they were assembled to generate a file in sbml-qual format (Supplementary Software 1) suitable for model exchange and computation[65].

**Simulation pipeline of the hybrid-cell scale model**. The workflow to combine the *R. solanacearum* GMI1000 VRN and the biochemical network[45] in one simulation pipeline first requires computing the regulatory network status under the considered environmental constraints. The gene status set by the regulatory network is then added as a constraint on the biochemical network in addition to environmental constraints. Then the flux balance analyses are performed (see Fig. 2). Each type of simulations was done as described below and done using the software FlexFlux[51]. The method used and described below is similar to the rFBA method[66] as both constrain flux bounds by regulatory network outputs. However, RSA does not require time-dependent simulations. Instead, the outputs of the regulatory network are computed by looking for an attractor in the network states. These network states can be considered as a steady state of the regulatory network (see ref. [51]).

The simulation of the regulatory network state was performed using the regulatory steady-state analysis (RSA) which is based on multi-state logical modeling[51]. Briefly, the network component status was assessed by computing synchronous update of the component status following their logical rules until an attractor was reached. The attractor, called steady-state, could be a point attractor or a cyclic attractor. The initial state of network components used to initiate the simulation were either (i) determined from the environment, (ii) set to 0 in case of gene KO mutants or internal network component disruption analysis, (iii) set using the default network component state (see Supplementary Note 1). If an attractor is found, the arithmetic mean value of each component through the attractor was computed and used as network status value determining the constraints for subsequent biochemical network computation.

Simulation of the biochemical reaction network state containing the genome-scale metabolic network module plus the macromolecule network module was performed using constraint-based modeling. Flux distributions were simulated by FBA and lower and upper values of the fluxes were determined using Flux Variability Analysis. Various constraints were applied on simulations: i) environmental conditions were defined by constraining lower and/or upper bound of exchange fluxes depending on the availability of the metabolite/macromolecule, or experimentally measured fluxes, ii) gene KO mutants or internal network component disruption were simulated by setting flux (or the gene involved in a GPR) value to 0, iii) the regulatory network constraints were taken into account by setting a gene involved in a GPR or reaction flux (lower bound and upper bound) to appropriate value depending on the regulatory network states solved previously. A broad diversity of phenotypes were simulated depending on the biological function (reaction flux) optimized in FBA.

**Biological entity clustering by contribution to the objective function**. The Biological Entity Clustering (BECO) method has been designed to classify the biological entities (reactions, genes, regulators) depending on their contribution to the optimization of different cellular objectives in different conditions. For each pair condition/objective, a classification of the metabolic genes depending on their contribution to the objective function (see supplementary Note 5 for details) was performed. The classification used in BECO is inspired by the gene classification proposed by Lewis et al.[27] The gene categories depending on their contribution to the optimality of the objective function value are (see Fig. 5):

ess: essential gene whose inactivation makes null the objective function value.
red: redundant gene associated to an essential reaction which is associated to other genes.
opt: optimal gene which is not essential but required for the optimality of the objective function value.
ele: less efficient enzymatically gene whose the activation implies a number of activated enzymes (reactions) greater than the absolute minimum number of enzymes required for objective function optimality.

The same categories (with the suffix _c) are extended to condition-dependent categories. For instance, a gene in the category ess-c is essential for the objective function only in specific conditions.

**Model performance in predicting phenotypes**. Qualitative evaluations of the cell scale model performance in predicting the bacterial phenotypes were performed by comparing the model predictions with experimental measurements. Phenotype predictions in various environments were performed by in silico simulation as described in the previous section by optimizing the biochemical reaction corresponding to the investigated phenotype. If the reaction flux > $\alpha$ ($\alpha = 0$) the phenotype is considered as performed by the bacteria and thus the prediction is classified as active phenotype (1), or if the reaction flux ≤ $\alpha$ the prediction is classified as not active phenotype (0). Then a binary matrix ($i \times j$) containing the

contingency of the experimental phenotype (i) and network prediction phenotypes (j) with the two classes (0, 1) are built and the network performance metrics are calculated as following. The true positive (TP) phenotypes correspond to identical value of 1 for i and j. The false positive (FP) phenotypes correspond to a value of 0 for i and 1 for j. The true negative (TN) phenotypes correspond to identical value of 0 for i and j, and the false negative (FN) phenotypes correspond to a value of 1 for i and 0 for j. The sensitivity (Sn) of the model prediction capacity, which corresponds to the true positive rate is calculated as following, where (P) is the number of experimentally active results ($i = 1$):

$$Sn = \frac{TP}{P}.$$

The precision (Pr) which is the positive predictive value is calculated as following:

$$Pr = \frac{TP}{(TP + FP)}.$$

The faux positive rate (FPR) is calculated as following:

$$FPR = \frac{FP}{N}.$$

The accuracy (Acc) of the model prediction capacity is calculated as following:

$$Acc = \frac{(TP + TN)}{(P + N)}.$$

**Model performance in predicting differential gene expression**. Evaluation of the VRN performance in predicting differential gene expression was done by comparing the model prediction with the experimental transcriptomic data-set produced by Jacobs et al.[50] For the in silico analysis, state of the regulatory network was predicted following the RSA method described above in different environments (see Supplementary Data 2 for the list and composition). Differences in gene states predicted between one tested environment and the complete medium used as reference in Jacobs et al.[50] were calculated. Then, genes were classified in three discrete classes (k), differentially repressed (−1), not changed (0), activated (1). For the experimental data analysis, transcriptomics data were discretized in three similar classes (l) (−1, 0, 1) considering differential gene expression by using a threshold of 2, i.e., twice repressed or twice more expressed in planta vs complete medium. Only genes harboring a differential expression with a p-value <0.01 were considered. Then, metrics of the model prediction were computed for each class. Subsequently the global mean prediction (M) was calculated by averaging the metrics of the three classes, see Supplementary Note 2 for details calculation. Briefly, performance of the model prediction was assessed by calculating F1 score (mean) (F1(M)).

$$F1_{(M)} = \frac{2 \cdot Pr_{(M)} \cdot Sn_{(M)}}{Pr_{(M)} + Sn_{(M)}}.$$

Pr(M) is the mean precision calculated as the average precision of the prediction for the three classes, where n is the number of classes:

$$Pr_{(M)} = \frac{\sum^{P} r_{(j)}}{n}.$$

Sn(M) is the mean sensitivity, or true positive rate, calculated as the average sensitivity of the network prediction for the three classes:

$$Sn_{(M)} = \frac{\sum^{S} n_{(j)}}{n}.$$

The sensitivity of network prediction to environmental changes, i.e., the deviation of the network prediction due to false assignment of environmental conditions or environmental perturbation was assessed as following. We computed deviation of the network performance due to environment parameters by randomly modifying one of them and comparing the obtained network performance to the one obtained in the original environment.

Detail of the algorithms for in silico analyses can be found in the Supplementary Note 2; scripts are freely available and can be downloaded at the following location (http://lipm-bioinfo.toulouse.inra.fr/systemsbiology/models/rsolanacearum).

**Phylogeny analyses**. Homology groups of proteins sequences in 29 bacterial genomes downloaded on NCBI database (see Supplementary Table 2 for the list) were assigned using OrthoMCL[67] via Family-Companion (http://family-companion.toulouse.inra.fr). Sequence clustering was performed using a p-value cut-off of $1 \times 10^{-5}$, and a percent match cut-off of 80. Sequences were aligned with Mafft v7.271[68]. Columns with more than 50% of gaps were removed. Phylogenetic trees were computed with IQ-tree (automatic mode)[69] and drawn with Itol (https://itol.embl.de/).

**Code availability**. The BECO method is available in the FlexFlux library (http://lipm-bioinfo.toulouse.inra.fr/flexflux/documentation.html#beco). Codes for VRN validation are available from the corresponding author upon request.

**Data availability**. The data that support the findings of this study are available in the website dedicated to the cell model of Ralstonia solanacearum (http://lipm-bioinfo.toulouse.inra.fr/systemsbiology/models/rsolanacearum/download.html). The metabolic network is available in BioModels[70] with the identifier MODEL1612020000. The VRN model was deposited also in BioModels and assigned the identifier MODEL1710170000.

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

## Acknowledgements

R.P. was supported by EMBO (Long-Term Fellowship ALTF 1627–2011) and Marie Curie Actions (EMBOCOFUND2010, GA-2010-267146) and European Research Council (ERC-StG336808 project VariWhim). We thank Christian Boucher and Patrick Barberis for their contribution to the construction and screen of the Tn5 insertion mutant library. We acknowledge funding from the Institut National de la Recherche Agronomique (Plant Health Division grant AAP SPE 2012) and the French Laboratory of Excellence project TULIP (ANR-10-LABX-41; ANR-11-IDEX-0002-02).

## Author contributions

R.P., L.C. and S.G. conceived and designed the study. R.P. reconstructed the VRN model and performed the experimental validation of model predictions. R.P. and L.C. conducted simulations and computational analyses. R.P., L.C. and L.M. conceived algorithms. L.C. and L.M. developed bioinformatic tools and the website. R.P., L.C. and S.G. wrote the manuscript.

## Additional information

**Competing interests:** The authors declare no competing financial interests.

