## [Peer Review File · Nature Communications]

Reviewers' comments:

Reviewer #1 (Remarks to the Author):

To help understand the robustness of pathogenic bacteria to their hosts, Peyraud et al. developed an integrated model of the metabolism and metabolic regulation of the bacterial plant pathogen *Ralstonia solanacearum*. The model includes a genome-scale FBA submodel of *Ralstonia solanacearum* metabolism and a logical submodel of the regulation of its metabolism by extracellular signals and nutrients, signaling pathways, and transcription factors.

After constructing and validating the model, Peyraud et al. simulated the model under a range of perturbations and analyzed the simulation results to find that *Ralstonia solanacearum*'s simulated housekeeping phenotypes are more robust to perturbations than less essential virulence-associated phenotypes. Furthermore, Peyraud et al. conducted a comparative sequence analysis of the VRN-regulated housekeeping genes, finding that the VRN-regulated genes are evolutionarily new, suggesting that *Ralstonia solanacearum* has recently acquired additional housekeeping genes to gain more robustness to interactions with its host.

The manuscript presents a new regulatory model of *Ralstonia solanacearum* and the manuscript presents an interesting and new application of regulatory flux balance analysis to the study of robustness and infection, and the manuscript contributes new understanding on how *Ralstonia solanacearum* has evolved to become pathogenic.

However, as described below, we have two concerns about the methodology underpinning the conclusions presented in the manuscript, as well as several minor suggestions for ways to enhance the clarity of the manuscript.

Major concerns

=====

Because much of the conclusions depend on the completeness of the reconstructed VRN, further analysis is needed to rule out the possibility that the conclusions are artifacts of the VRN reconstruction that might have missed true regulations of housekeeping functions. This could be addressed, for example, by comparing how much the housekeeping phenotypes are regulated by the VRN in comparison to the rest of the model and by measuring the variation of the housekeeping phenotypes, as compared to the virulence-associated phenotypes, in planta. Similarly, how does the genetic and functional redundancy of the housekeeping and virulence-associated phenotypes compare?

The abstract states the virulence network has evolved to promote robustness upon infection. However, the manuscript only briefly explores infection. Additional simulations are needed to support this claim. For example, the simulations described in lines 210-285 could be compared against simulations of conditions that represent infection.

Minor concerns

=====

The following concerns are intended to suggest ways to clarify the manuscript:

Line 95: To further emphasize the novelty of this work, we think it would be helpful to briefly summarize other efforts to model pathogens and their interactions with their hosts.

Figure 1:

- The three major types of robustness mechanisms could be highlighted using three colors
- The type of robustness would be clarified by using "versatility" and "plasticity" consistently

Lines 115-123: To make it easier to understand the methodology, it would be helpful to cite regulatory flux balance analysis (rFBA) and/or compare the presented approach with rFBA.

Please describe the macromolecule network (line 117). What does this represent?

Figure 2C: Please increase the sizes of the markers

Please elaborate on the results of the validation of the VRN around line 152

Please elaborate on the results of the validation of the hybrid model around line 165-167

To make it easier to understand the methodology, throughout the Results section, it would be helpful to explicitly point out which steps served as controls.

Table S3: Please correct the supplementary material number in the Table of Contents

Please provide more detailed captions for the supplementary tables. For example, please explain the meaning of each of the columns in Table S3. How do these columns specify the validation simulations? Which conditions were predicted correctly and incorrectly?

Please correct minor grammatical errors throughout the manuscript, e.g.

- Lines 45, 48: use consistent italicization for "i.e."
- Line 60: Add dash for compound word "network[-]scale"
- Line 70, 72: add missing commas "... genetic redundancy[,], but their extent ..." and "... within biological systems[,], but that modulation ..."

Please deposit the model to the BioModels repository

Reviewer #2 (Remarks to the Author):

I commend the authors for examining a very interesting scientific question. They have conducted a detailed examination. I also like the new metric they introduce to report robustness of a system. However, I find that they have been grossly negligent in describing their methods in the main text and supplementary materials. Overall, I have a number of concerns:

Major comments

1. The authors need to give an idea of how complete is their VRN. For example, what fraction of genes in the genome of *R. solanacearum* do not have any associated function? Could the fact that their regulatory modules are dominated by interactions associated with metabolism and biomass production be an artifact of scientists' early preference for identifying these interactions? How would that effect their overall predictions?
2. In a similar vein, why work on *R. solanacearum* and not an organism such as *E. coli* or other model pathogens for which the VRN has been examined in much more detail and hence the generated VRN would be more complete?
3. With regards to the validation of gene expression predictability: while the authors proved a nice figure describing their method and provide a list of conditions that were examined, they do not prove any raw data or statistical results for their conclusion that the model "reliably predicted the transcriptional responses controlled by VRN". Also, they note 4 key environmental constraints acting on the VRN but do not proceed any further with this information.
4. For validation of phenotype predictability, table S3 has only one column detailing a phenotypic outcome. It is unclear whether this is the model prediction or the experimental observation. Also,

in order to assess the accuracy of predictions, it is necessary to have another column which lists the theoretical/experimental results (whichever is not listed in column G).

5. Testing for contribution of each entity to a phenotype, the classification criteria should be detailed in the supplementary materials. The reference to Wang and Lewis et al does not suffice.

6. I cannot find Figs S2 or S3 in the submitted documents.

7. I'm confused by the paragraph starting on line 276. The authors claim they conducted a BECO analysis to identify the conditions that mobilized the VRN-regulated genes. In the next sentence, they claim that one of the three conditions is an environment not activating VRN.

8. The authors need to give a cursory description of methods they have used such as orthoMCL. At times, this paper reads like an internal memo among members of a research team, instead of a manuscript to be shared with a wide audience.

9. The authors only examined the VRN control of Amino acid biosynthesis pathway when assessing the evolutionary origin of VRN controlled genes. There are other pathways that are broadly shared among organisms, such as glycolysis, pentose phosphate pathway, and purine biosynthesis pathway. Given the author's generalized assertion that "VRN plugs into the primary metabolism mainly through the control of genes likely acquired via horizontal gene transfer", they need to examine more than one metabolic pathway.

10. Visually examining Figure S9, the bias in GC content between the regulated and unregulated genes seems negligible (~1%). It would be useful for the authors to either provide the raw data or show how they calculated the reported p-value.

11. The authors do not provide any detail about how the constraint-based model of metabolism in *R. solanacearum* was generated. Is this a curated model or a draft model? Were the FBA model's predictions validated against any experimental data? This information is crucial for judging the validity of the model's predictions.

12. The FlexFlux method uses the average of the upper and lower bounds of all the intervals to form a steady-state constraint. Isn't there a possibility that such constraints actually might preclude the system from moving to the predicted attractors?

13. The authors do not provide enough detail about the FlexFlux methodology in the paper or the supplementary materials. This information is vital for a facile understanding of the research conducted and reported results.

Minor comments:

1. The language used for sentences that form lines 80-87 is awkward. This could be due to my familiarity to American English as opposed to British English. However, terms such as "as regard to" or "patterns which carry robustness" are hard to follow. I'm more familiar with "with regard to" and "patterns which might affect robustness".

2. Supplementary table 3 has a header that reads Table 7

3. Supplementary material 2, pg 3, line 7, in front of an internal perturbation

4. Need to reference which *E. coli* and *P. aeruginosa* models were used for the phenotype analyses.

5. What fraction of the genes in each model are the 421 orthologs?

6. What biomass differences account for the differences in the robustness between the three examined organisms?

7. Figure 1 legend: environmental perturbation: quantitative or qualitative changes.

8. Figure 3A, for figure for the components by subsystems, it is very hard to link some titles to the associated line.

9. Figure 5B: recommend moving the terms like Essential and Redundant of essential above biochemical figures to ease the process of reader linking the terms to shorthands like ESS and RED.

Reviewer #3 (Remarks to the Author):

Review of Peyraud, Cottret, Marmiesse and Genin: "Control of primary metabolism by a virulence regulatory network promotes phenotypic robustness in a bacterial plant pathogen."

Overview:

In this manuscript, the authors couple an inferred regulatory network (VRN) to a genome-scale metabolic model from the plant pathogen *R. solanacearum*. They then model the ability of this hybrid model to respond to environmental and internal perturbations to assess the nature of the robustness of this organism with respect to its interaction with its plant host.

They find that the VRN interacts with only part of the metabolic network and that this interaction is biased away from core-housekeeping genes. Many of the genes controlled by the VRN are functionally redundant with other genes in the genome (e.g., Figure 6) and many of these redundant genes appear to owe their origins to horizontal gene transfer.

Major comments:

This is an ambitious and important paper, with the inference of the VRN and its coupling to the metabolic network being an elegant addition to our knowledge. The discovery of the horizontal transfer of parts of metabolism into this organism to allow virulence-related regulation is, to my knowledge, new and quite interesting.

Nonetheless, I have some concerns with the presentation of the work. First, the authors need to be more careful when they equate biological robustness and inferred robustness from their computational models. For instance, in the caption of Figure 4, they imply that the VRN restricts robustness. But what is actually being seen is different, I believe. Genome-scale metabolic models (GSMM) have a well-known issue with inferring that all genetic redundancy in enzymes represents robustness. In other words, a GSMM will always interpret a second copy of an enzyme as conferring robustness. But of course in real biological systems, these redundant enzymes are rarely able to perfectly complement each other: they may be expressed at different times or locations, be regulated differently etc. So what the authors actually see in Figure 4 is that adding regulatory information to a GSMM gives robustness inferences that are likely closer to the underlying biological reality than is the GSMM alone. Thus, since the authors did not infer a general regulatory network, the higher robustness of the housekeeping genes is likely an artifact of the absence of regulatory information here. Overall, the manuscript makes clear that the authors are aware of this issue, but the wording is not as careful as it could be in a number of places. See especially lines 223-232 and 281-284 for more instances of this issue.

On lines 257-261, the authors claim that the VRN changes estimated robustness most strongly for "proliferation." But since for the cell to proliferate a very large number of pathways must function correctly, it would seem obvious that this phenotype would be the more sensitive than phenotypes impacting a smaller number of pathways. Now, I may well misunderstand what the authors mean here, but in that case, this section needs clarification.

Likewise, I almost could not make heads or tails of lines 262-275—this section needs heavy rewriting.

I think one of the reasons I had difficulties with several of these sections is that the authors, in an attempt to make the manuscript accessible, have used common terms like "phenotype" and "environmental perturbation" to refer to particular features of their model. However, one has to dig deeply into the supplements to find the actual model definitions of these abstractions. The authors need to clarify and give examples: for instance a phenotype might mean the ability to produce a certain compound on a certain media according to their model. Likewise, "environmental perturbation" calls to mind a quantitative change in the environment (lower pH, higher temperature, reduced concentration of some nutrient or hormone). However, GSMM are notorious for not being able to model such changes well. So many of the "perturbations" are going to be more like presence/absence tests—loss of a nutrient, gene deletion and so forth. The authors need to make the reader aware of this distinction.

Minor comments:

“Ortholog” is misused in Figure 6: either a gene is or is not an ortholog of a particular gene (e.g., had a common ancestor at the last common ancestral species or did not).

Response to Referees

We really appreciated the constructive comments and recommendations made by the reviewers. We have taken into consideration the different concerns and remarks, and have corrected our text accordingly. We also paid attention to the recommendation to better define some of the terms and methodology used (see Answers#20, 23, 28 and #33) and weighted some of the statements as suggested by Reviewer# 3 (see Answers#30).

Please find hereunder our point-to-point responses (in blue in the text below) to the reviewer's comments (in black). You will find the changes tracked in the resubmission document. A clean copy of the ms is also joined.

Reviewer #1

1. "Because much of the conclusions depend on the completeness of the reconstructed VRN, further analysis is needed to rule out the possibility that the conclusions are artifacts of the VRN reconstruction that might have missed true regulations of housekeeping functions. This could be addressed, for example, by comparing how much the housekeeping phenotypes are regulated by the VRN in comparison to the rest of the model and by measuring the variation of the housekeeping phenotypes, as compared to the virulence-associated phenotypes, in planta."

The point raised about completeness of the VRN is important to consider and we share this concern about potential bias due to incompleteness of models. We present below arguments and novel analyses to justify why we believe that our reconstructed model is significant, both in terms of quality and in terms of completeness.

First, we would like to recall that we performed extensive experimental validations using large-scale analyses: transcriptomic reprogramming based on experimental data, phenotyping of 15 single and multiple mutants in VRN components, creation of a Tn5 insertion library used for essentiality gene screening. In these three cases, we tested the model behavior against these experimental datasets that were not used for the VRN reconstruction. In addition, we would like to highlight the level of validation we also performed in our previous study for the metabolic network (including metabolic flux analysis and and Biolog screen) (Peyraud et al., PLoS Pathog 2016). To our point of view, the capability of the model to predict a phenotype is the best criterion to assess its completeness. However, as requested by the reviewers, we provide in the following paragraphs other metrics to estimate this completeness. Note that none broadly accepted metrics exists yet on which we could stand.

Second, we agree with both reviewers#1 and #2 that some additional metrics could allow an estimation of the completeness of the model. As requested, we have performed the following analyses:

a) Extent of genes included in the network. We calculated the proportion of genes included in the model (reaction network and VRN) on the total number of *R. solanacearum* genes assigned to a functional class in the reference MAGE annotation (<http://www.genoscope.cns.fr/agc/microscope/home/index.php>). Among those functional categories, the enzyme category is a good readout to assess the extent of the metabolic network. Among the 995 genes in this category, 707 (71.1 %) are included in the reaction network. The remaining 28.9 % mainly correspond to enzymes involved in cellular signaling (e.g. kinases, phosphatases...) or processes (DNA or RNA -dependent processes, transposases...) but which are not

directly involved in small molecules metabolism. The same analysis was done for the regulator category: among the 140 genes assigned by MAGE in this class, 48 (34.3%) genes are included in the VRN. Also included in the VRN are 9 out of the 12 genes assigned to the receptor category and 74 out of the 82 genes annotated in the virulence-associated phenotype category. This level appears to us as a significant level of completeness for the VRN since the regulator category also includes regulatory genes not directly associated to virulence (e.g stress response regulators or some metabolic regulators). Details about the number and proportion of genes in the genome annotation categories are provided in the supplementary table S2.

We added this analysis to the manuscript:

lines 139 – 143. “The resulting hybrid model includes a substantial proportion of genes annotated as enzymes (71.1%), transporters (65.3%), regulators (34.3%), receptors (75.0%) and virulence-genes (90.2%), indicating a significant level of completeness of the model in term of contemporary knowledge about metabolism and virulence regulation (Supplementary table S2)”

b) Extent of genes regulated by the VRN. Among the 995 and 613 genes annotated to code for enzymes and putative enzymes in the genome, 85 (8.5%) and 64 (10.4%), respectively, were found to be included in the VRN. They represent 21.0% of the genes included in the VRN. The gene classes the most enriched in genes included in the VRN are those contributing to virulence-associated phenotypes (90.2%), receptors (75.0%), regulators (34.3%), or structural components (33.0%). Hence, this analysis shows that there is no bias toward the control of metabolic functions by the VRN. Conversely, considering the high proportion of metabolic genes in the genome, it seems to indicate that the control on the metabolism is less important compared to some others functions.

Information about how the VRN-regulated genes were separated into annotation categories is now available in the supplementary table S2.

c) Extent of unknown components in *R. solanacearum*. 1784 out of the 5708 genes of the *R. solanacearum* genome (31.6%) are ORF of unknown function. This is approximately twice the level of *E. coli* K12 (14.65%) (MAGE source). If considering *Pseudomonas aeruginosa* PAO1, which is also a well-studied pathogenic bacterium with a genome size comparable to *R. solanacearum*, there are 2038 (34.5%) unknown proteins for a total of 5913 genes. The proportion of genes of unknown functions is therefore comparable in *P. aeruginosa* and *R. solanacearum* (and in the range of the proportion estimated in many other bacteria with a similar genome size). In this regard, we would like to mention that the genome annotation quality is a key driver of the model reconstruction quality. Annotation of *R. solanacearum* strain GMI1000 was not performed automatically but instead involved expert manual annotation (Salanoubat et al., Nature 2002) followed by a second round a genome annotation/curation in the beginning of the 2010's (Peyraud et al., Plos Path. 2016).

d) We compared the size of the VRN we reconstructed with the VRN of *P. aeruginosa*, one of the most extensive regulatory network described for a bacterial pathogen. The size of the *R. solanacearum* VRN is comparable to the *P. aeruginosa* regulatory network reconstructed so far: the model reconstructed by Galán-Vásquez et al. (Microb Inform Exp. 2011) contains 1020 regulatory interactions and 650 genes whereas, the *R. solanacearum* VRN reconstructed in our study possesses 1443 regulatory interactions and 712 genes.

We added this information in the manuscript:

lines 380-382. “Thus, this is one of the most complex VRNs reconstructed so far in bacterial pathogens^{43,62}, comparable in complexity to the reconstructed *P. aeruginosa* VRN³⁶”

Third, we conducted an analysis of the phenotypic regulation by the VRN as suggested by reviewers in order to reveal if a bias in the network prediction may be indicative of its incompleteness:

e) Differential regulation of housekeeping and virulence phenotypes. One concern raised by the reviewer was the possibility of bias dependent on the completeness of the reconstructed VRN. As

requested, we assessed if the number of VRN-regulated genes involved in housekeeping phenotypes was significantly not different from the number involved in the virulence-associated phenotypes. On the set of 16 phenotypes investigated in the study, we found that 14.3 ± 2.6 % of VRN-regulated genes for the 6 housekeeping phenotypes and 15.1 ± 4.5 % VRN-regulated genes for the 10 virulence phenotypes. This showed that the difference for the set of phenotypes investigated in this study was not significant, t-test p-value 0.628.

We added this analysis into the manuscript:

lines 244-246. “This difference was not due to a difference between the number of VRN-regulated genes involved in the housekeeping (14.3 ± 2.6 %) and virulence phenotypes (15.1 ± 4.5 %), p-value 0.628.”

f) Regulation of housekeeping and virulence phenotypes *in planta*.

We performed the comparison suggested by reviewer 1 concerning *in planta* regulation of housekeeping and virulence associated phenotypes. We compared the robustness of housekeeping and virulence functions, using simulations for *in vitro* and *in planta* environments with regard to environmental perturbations. We found a higher robustness of the virulence functions *in planta* (0.741 ± 0.313) compared to *in vitro* (0.208 ± 0.140) conditions (t-test p value: 0.0006). The robustness of the housekeeping functions however remained identical (1.00) upon environmental perturbations *in planta* as well as *in vitro*.

We have added the result of this analysis in the manuscript:

lines 257-259. “When comparing conditions that activate virulence functions, the robustness of the virulence phenotypes is higher *in planta* (0.741 ± 0.313) compared to *in vitro* conditions (0.208 ± 0.140), p-value 0.0006.”

2. Similarly, how does the genetic and functional redundancy of the housekeeping and virulence-associated phenotypes compare?

We conducted the analysis suggested by the reviewer. To do so, we used the genes classification done by the BECO analysis for the analysis of genetic redundancy (RED) or functional redundancy (OPT, ELE) (Figure 5 and Suppl. Material Table S9). We did not find a significant difference in the proportion of genes involved in housekeeping and virulence phenotypes. t-test p-values: ESS 0,990; RED: 0,862; OPT 0,860; ELE 0,442; ESS-C 0,110; RED-C 0,088; OPT-C 0,608; ELE-C 0,759.

We have added the result of the BECO analysis in an additional supplementary table S9.

3. The abstract states the virulence network has evolved to promote robustness upon infection. However, the manuscript only briefly explores infection. Additional simulations are needed to support this claim. For example, the simulations described in lines 210-285 could be compared against simulations of conditions that represent infection.

We stated in the abstract that our results ‘support the view’ that the VRN ‘may have evolved’ to promote robustness *in planta*. To assess further this point, we performed the analysis proposed by the reviewer.

a. Robustness with regard to environmental perturbations.

The variation of robustness with regard to environmental perturbations is presented above in Answer#1f. This analysis showed that the robustness of the virulence-associated phenotypes is higher *in planta* compared to *in vitro* (p-value 0.0006).

b. Robustness with regard to internal perturbations.

We compared the variation of robustness with regard to internal perturbations between *in planta* and *in vitro* conditions. For each environment, an internal perturbation corresponds to a single *in silico* knock-out (loss of function) for a metabolic gene, or a regulator, or a reaction (see BECO

analysis in Material and Methods). The simulations showed that the robustness of the proliferation phenotype is significantly higher *in planta* compared to *in vitro* (p-value 0.0005): 0.751 ± 0.016 (for 8 environments tested) *in vitro* versus 0.783 ± 0.004 (for 6 environments tested) *in planta*. All the phenotypes displayed an increase of robustness *in planta*. For four of them (proliferation, swimming motility, pectin methyl-esterase secretion and PHB storage), we could establish that the difference between *in planta* and *in vitro* conditions was statistically significant because there were enough environmental conditions in which the corresponding pathways were activated. For instance, robustness of the pectin methyl-esterase secretion is 0.771 ± 0.005 *in vitro* versus 0.797 ± 0.006 *in planta* (p-value 0.0032).

We added this analysis in the manuscript:

lines 310-314. "This analysis also suggested that phenotypic robustness is higher *in planta* compared to *in vitro* conditions. We conducted simulations to test the robustness of the proliferation function in different environments: this indeed revealed that robustness was significantly higher *in planta* (0.783 ± 0.004 , for 6 environments tested) compared to *in vitro* (0.751 ± 0.016 , for 8 environments tested), t-test p-value 0.0005."

4. Line 95: To further emphasize the novelty of this work, we think it would be helpful to briefly summarize other efforts to model pathogens and their interactions with their hosts.

We added in the introduction the following sentence:

lines 93-95. "Reconstruction of metabolic models of pathogens and studies of their robustness with regard to internal perturbations were conducted with the aim to discover new drug targets (Chavali et al., 2012; Rienksma et al 2014) but only few virulence regulatory networks were reconstructed so far (Galán-Vásquez et al., 2011)".

5. Figure 1:

- The three major types of robustness mechanisms could be highlighted using three colors
- The type of robustness would be clarified by using "versatility" and "plasticity" consistently

We modified the figure accordingly to the reviewer's recommendations.

6. Lines 115-123: To make it easier to understand the methodology, it would be helpful to cite regulatory flux balance analysis (rFBA) and/or compare the presented approach with rFBA.

We agree that the FlexFlux method is similar to the rFBA approach. The main difference is that the FlexFlux method does not require time-dependent simulations since it searches for attractors in the regulatory network. This approach allowed to simulate all single mutant phenotypes for each conditions (14) and each phenotype (16) (>300 000 simulations solved in 2 days of computation time), and has therefore enabled to perform the BECO analysis.

We have added the following sentences in the text:

lines 481-485. "The method used and described below is similar to the rFBA method (Covert et al., 2008) since both constrain flux bounds by regulatory network outputs. However, RSA does not require time-dependent simulations. Instead, the outputs of the regulatory network are computed by looking for an attractor in the network states. These network states can be considered as a steady state of the regulatory network (see Marmiesse et al., 2015)."

7. Please describe the macromolecule network (line 117). What does this represent?

The macromolecule network corresponds to the reactions required for synthesis, transport and association of macromolecules like proteins. Because *R. solanacearum* is a pathogen known to produce a large array of proteinaceous virulence factors, this macromolecule network was created in a former study aimed to estimate the metabolic cost for such virulence factor biosynthesis (Peyraud

et al, PLoS Pathog 2016). For instance, synthesis of the plant cell wall-degrading pectin methyl-esterase enzyme (PME) and its secretion cannot be formally included in the metabolic network. To differentiate such type of reaction associated to macromolecules from metabolic reactions, we grouped them in a so-called “macromolecule network”. Because both metabolic and macromolecule networks are biochemical reaction networks, they were merged in one SBML file to perform FBA. Hence, optimal fluxes distribution for PME secretion (cost of secretion + biosynthesis from amino acids + cost of amino acids synthesis) can be calculated in one FBA simulation. For details on the reconstruction of the metabolic plus the macromolecule network, see Peyraud et al, PLoS Pathog 2016.

8. Figure 2C: Please increase the sizes of the markers

We think that the Figure 3C should be the one to be modified instead of the 2C since there is no 2C and it make sense to increase the markers sizes in the 3C. We modified it accordingly.

9. Please elaborate on the results of the validation of the VRN around line 152

& 10. Please elaborate on the results of the validation of the hybrid model around line 165-167

As stated above, extensive experimental validations were performed to assess the validation of the reconstructed models (VRN and metabolic network). It included transcriptomic reprogramming based on experimental data, phenotyping of 15 single and multiple mutants in VRN components, creation of a Tn5 insertion library used for essentiality gene screening. The model behavior was tested against experimental datasets that were not used for the VRN reconstruction. Since we had constraints on the size of the text, further explanations on the validation of the reconstructed models are provided in the supplementary material 1.

11. To make it easier to understand the methodology, throughout the Results section, it would be helpful to explicitly point out which steps served as controls.

It is not clear for us to what kind of controls the reviewer refers. The metrics provided, i.e. accuracy and sensitivity, are informative of the predicting capacity of a model. We revised the text in the Results section in many places and hope that the reading is now more comprehensive.

12. Table S3: Please correct the supplementary material number in the Table of Contents

We modified the Table of Contents accordingly.

13. Please provide more detailed captions for the supplementary tables. For example, please explain the meaning of each of the columns in Table S3. How do these columns specify the validation simulations? Which conditions were predicted correctly and incorrectly?

Table 3 was entirely revised to address this comment. The table number is 5 in the newly submitted manuscript.

14. Please correct minor grammatical errors throughout the manuscript, e.g.

- Lines 45, 48: use consistent italicization for "i.e."
- Line 60: Add dash for compound word "network[-]scale"
- Line 70, 72: add missing commas "... genetic redundancy[,] but their extent ..." and "... within biological systems[,] but that modulation ..."

Done.

15. Please deposit the model to the BioModels repository

The metabolic network model had already been added in the BioModels repository:

<https://www.ebi.ac.uk/biomodels-main/MODEL1612020000> and we have also added the virulence regulatory network: <https://www.ebi.ac.uk/biomodels-main/MODEL1710170000>

We have added the model ids in the text (lines 473-474, with reference#71).

Reviewer #2

16. The authors need to give an idea of how complete is their VRN. For example, what fraction of genes in the genome of *R. solanacearum* do not have any associated function? Could the fact that their regulatory modules are dominated by interactions associated with metabolism and biomass production be an artifact of scientists' early preference for identifying these interactions? How would that effect their overall predictions?

This point was also raised by reviewer#1, see response to point 1.

17. In a similar vein, why work on *R. solanacearum* and not an organism such as *E. coli* or other model pathogens for which the VRN has been examined in much more detail and hence the generated VRN would be more complete?

We agree that *E. coli* is still the reference model for bacteria in terms of global knowledge of the organism. The aim of our work was to initiate a system's biology approach on a model plant pathogen since these organisms have many specific features in terms of pathogenicity determinants compared to animal or human pathogens. *Ralstonia solanacearum* is recognized as one of the model bacterial systems in plant pathology: the complete genome sequence was the second established for a plant pathogen (Salanoubat et al., Nature 2002) and many functional studies have been conducted on this microorganism. For example, more than fifty regulatory components involved in virulence have been functionally characterized over the three last decades (see Genin & Denny, Ann. Rev. Phytopathol. 2012). This explains why the confidence score of the *R. solanacearum* VRN is 2.72 on the scale from 0 to 4 since experimental data (biochemical and genetic data) were the main sources of information used for the reconstruction. Compared to another model bacterial pathogen (*P. aeruginosa*), the completeness of the VRN reconstructed in *R. solanacearum* is similar in size, see response to point 1 paragraph c. We think that this study will pave the way for integrative studies carried out on other plant pathogenic organisms, and should provide a basis to develop novel control strategies against this major plant pathogen worldwide.

18. With regards to the validation of gene expression predictability: while the authors proved a nice figure describing their method and provide a list of conditions that were examined, they do not prove any raw data or statistical results for their conclusion that the model "reliably predicted the transcriptional responses controlled by VRN". Also, they note 4 key environmental constraints acting on the VRN but do not proceed any further with this information.

We performed an accuracy analysis to assess the performance of the model in predicting gene expression. This statistical method allows analyzing the performance of a predictor when a discrimination parameter varies and is therefore suitable to test the prediction capacity of the model when environmental conditions vary. We added the raw data to the supplementary data in addition to the accuracy, sensitivity, precision, F1 score of the model in predicting *in planta* gene expression (supplementary table S4). We also added the F1 score value in the text to present this data (line 157). The four major environmental constraints (*i.e.* plant cell wall sensing, quorum sensing, O₂ limitation and nitrate availability) are interesting since they correspond to key environmental factors in xylem

tissues in which bacterial multiplication is optimal. We did not focus further on this aspect since our objective in this study was the control of phenotypic robustness.

19. For validation of phenotype predictability, table S3 has only one column detailing a phenotypic outcome. It is unclear whether this is the model prediction or the experimental observation. Also, in order to assess the accuracy of predictions, it is necessary to have another column which lists the theoretical/experimental results (whichever is not listed in column G).

Table 3 was entirely revised to address this comment. The table number is 5 in the newly submitted manuscript.

20. Testing for contribution of each entity to a phenotype, the classification criteria should be detailed in the supplementary materials. The reference to Wang and Lewis et al does not suffice.

The gene and reaction classification methods are detailed in Supp. Material Table S9. For better understanding, we have added a description of each reaction class in the section “Classification of reactions and genes » in the supplementary materials.

“The reaction categories are:

- Essential reactions whose the deletion makes the objective function’s value to be null
- Zero flux reactions can’t carry flux, i.e. their flux value is always 0
- MLE (Metabolically less Efficient Reactions) enables objective function but not in an optimal way
- ELE (Enzymatically less Efficient Reactions) makes the objective function’s value to be optimal but by using more enzymes than other optimal solutions
- Objective Independent Reactions can carry flux but have no effect on the objective function
- pFBA optima reactions makes the objective function’s value to be optimal but by using a minimal number of enzymes”

21. I cannot find Figs S2 or S3 in the submitted documents.

We recast entirely the supplementary data, see supplementary material 1

22. I’m confused by the paragraph starting on line 276. The authors claim they conducted a BECO analysis to identify the conditions that mobilized the VRN-regulated genes. In the next sentence, they claim that one of the three conditions is an environment not activating VRN.

The environment that is not activating the VRN was used as control. We added this precision in the text (line 303). Also, we agree that the terms ‘activating / not activating VRN’ may be confusing. In fact, the term ‘activation’ was used to refer to an environment which is known (based on experimental data) to activate the virulence phenotype (and so presumably the VRN). However, it is important to keep in mind that the VRN does not only activate genes but also represses some. In this analysis, we tested if the VRN-regulated genes can be mobilized in a condition which is not activating the VRN. In such a case, upon stimuli activating the VRN these genes should be repressed, and such a repression could trigger a network optimization and a reduction of robustness. The result of this analysis (Fig S5) does not support this scenario, but instead indicates that robustness of the network increases upon stimulation.

23. The authors need to give a cursory description of methods they have used such as orthoMCL. At times, this paper reads like an internal memo among members of a research team, instead of a manuscript to be shared with a wide audience.

We apologize for this and tried to better detail several methodological aspects notably by providing the link of the tool we developed (see <http://family-companion.toulouse.inra.fr>) and used to run the analysis. The publication of this tool is under review; see the help section in <http://family-companion.toulouse.inra.fr> for the description of the workflow. However, orthoMCL parameters provided are sufficient to reproduce the results. Also, we provided an extensive supplementary data dedicated to the methods section in order to explained the methods we used with numerous illustration of workflow, see supplementary material 2.

24. The authors only examined the VRN control of Amino acid biosynthesis pathway when assessing the evolutionary origin of VRN controlled genes. There are other pathways that are broadly shared among organisms, such as glycolysis, pentose phosphate pathway, and purine biosynthesis pathway. Given the author's generalized assertion that "VRN plugs into the primary metabolism mainly through the control of genes likely acquired via horizontal gene transfer", they need to examine more than one metabolic pathway.

We agree with the reviewer that primary metabolism is not restricted to amino acid biosynthetic pathways. Our rationale to focus the analysis presented in the manuscript on these amino acids pathways is that they represent six independent pathways concentrating a high number of VRN-regulated genes (9) among the regulated genes in the primary metabolism (20). Only, few other primary metabolism pathways are regulated by the VRN. For instance, no gene involved in the glycolysis (Embden-Meyerhof-Parnas) is under the control of the VRN. However, to provide a more complete picture as requested by the reviewer, we included in the manuscript the analysis concerning the other VRN-regulated genes of the primary metabolism, see supplementary table S12 for detail. In total, 16 out of the 20 genes regulated are involved in functional or genetic redundancy, further supporting the hypothesis that the VRN plugs on genes providing robustness. Some of these additional genes were also likely acquired via horizontal gene transfer. These genes include a 2-oxoglutarate dehydrogenase e1 decarboxylase component (RSp1364) involved in the TCA cycle, a Glucose 1-dehydrogenase involved in the Entner-Doudoroff pathway (RSc0215), or a spermidine synthase (RSp0825) involved in polyamine biosynthesis. This information is now provided in lines 359-361 and in Supp. Material Table S12.

25. Visually examining Figure S9, the bias in GC content between the regulated and unregulated genes seems negligible (~1%). It would be useful for the authors to either provide the raw data or show how they calculated the reported p-value.

We agree that the difference seems marginal but the comparison involves a very large number of genes (305 VRN-regulated genes and 1122 VRN-independent genes), so the 1% difference is statistically significant (t-test p-value $3 \cdot 10^{-4}$; see Fig S9). The raw data are now provided in a supplementary table S11.

26. The authors do not provide any detail about how the constraint-based model of metabolism in *R. salancearum* was generated. Is this a curated model or a draft model? Were the FBA model's predictions validated against any experimental data? This information is crucial for judging the validity of the model's predictions.

We corrected the manuscript (lines 120-121) in order to state clearly that the genome-scale metabolic network and the macromolecule network were reconstructed in a previous study (Peyraud et al., PLoS Pathog 2016). The GSMN was validated qualitatively (Biolog and phenotype validation) and quantitatively (biomass quantification and metabolic flux analysis with the wild-type and a mutant strain defective for a master regulator of the VRN).

27. The FlexFlux method uses the average of the upper and lower bounds of all the intervals to form a steady-state constraint. Isn't there a possibility that such constraints actually might preclude the system from moving to the predicted attractors?

The reviewer pinpoints that the method used to calculate the regulatory constraints applied to the FBA may bias the finding of the attractor. First, we want to mention that the sequence of the algorithm performed by FlexFlux avoids missing the "expected" attractor at the regulatory level. It first finds the attractor within the regulatory network, and then computes the average of the states in the attractor in the case of a cyclic attractor. For nodes corresponding to reactions, each state found in the attractor is translated into lower and upper flux bounds. The final constraints on the reaction flux are defined by computing the average of these lower and upper flux bounds. At the end, the constraints are applied on the FBA. This calculation could interfere the finding of the attractors if there was time steps (as in rFBA) but it is not the case here: we computed first the regulatory network steady state and then we used the results for constraining the FBA. Hence, the potential bias due to the averaging methods does not interfere with the finding of the attractor within the regulatory network.

By computing the average of the upper and lower bounds, we wanted to consider an average metabolic behavior in a bacterial population. This means that state changes need to be important to change the average behavior of the population. By doing so, the weight of each sub-states (sub-population) is proportional to their proportion in the cyclic attractors. This assumption is suitable for the analysis we conducted because during plant infection *R. solanacearum* multiplies heavily, starting from hundreds of individuals to reach billions per gram of host tissue. However, we recognize that some bias can occur using this method in cases such as cell synchronization and fine temporal analysis, single cell analysis, or high heterogeneity in the host tissues considered.

28. The authors do not provide enough detail about the FlexFlux methodology in the paper or the supplementary materials. This information is vital for a facile understanding of the research conducted and reported results.

We apologize that the structure of the supplementary data was not suitable for easy understanding. On purpose to clarify the methodological information, we recast the supplementary material in one single file with organized table of content. The FlexFlux overall methodology as well as the BECO method are explained in Supplementary Material 1 in the method section.

29. Minor comments:

The language used for sentences that form lines 80-87 is awkward. This could be due to my familiarity to American English as opposed to British English. However, terms such as "as regard to" or "patterns which carry robustness" are hard to follow. I'm more familiar with "with regard to" and "patterns which might affect robustness".

Text in lines 80-87 has been corrected . Other proposed corrections were also done.

Supplementary table 3 has a header that reads Table 7

Corrected

Supplementary material 2, pg 3, line 7, in front of an internal perturbation

Corrected

What fraction of the genes in each model are the 421 orthologs?

There are 1366 genes in the *E. coli* model iJO1366 used, 1086 in the *P. aeruginosa* iMO1086 model used and 1476 in the *R. solanacearum* iRP1476 model used. Thus, the 421 genes represent 30.8%, 38.8% and 28.5%, respectively.

What biomass differences account for the differences in the robustness between the three examined organisms?

Mainly amino acid charging on tRNAs and specific biomass composition are responsible of the discrepancy.

We added the following information into the supplementary material file S1:

“The higher robustness observed for *E. coli* and *P. aeruginosa* was tracked and found to be mainly due to differences in the biomass equation, ranging from 0.04 to 0.06 of robustness. For instance, the main differences between *R. solanacearum* and *E. coli* are due to the amino acid charged on their tRNA which is taken into account into the *R. solanacearum* biomass equation but not in the *E. coli*’s one.”

Figure 1 legend: environmental perturbation: quantitative or qualitative changes.

Corrected. In the new version, we added the definition of perturbations in the main text (see lines 207-214)

Figure 3A, for figure for the components by subsystems, it is very hard to link some titles to the associated line.

Corrected. We tried to improve location of the labeling.

Figure 5B: recommend moving the terms like Essential and Redundant of essential above biochemical figures to ease the process of reader linking the terms to shorthands like ESS and RED.

Corrected

Reviewer #3

30. First, the authors need to be more careful when they equate biological robustness and inferred robustness from their computational models. For instance, in the caption of Figure 4, they imply that the VRN restricts robustness. But what is actually being seen is different, I believe. Genome-scale metabolic models (GSMM) have a well-known issue with inferring that all genetic redundancy in enzymes represents robustness. In other words, a GSMM will always interpret a second copy of an enzyme as conferring robustness. But of course in real biological systems, these redundant enzymes are rarely able to perfectly complement each other: they may be expressed at different times or locations, be regulated differently etc. So what the authors actually see in Figure 4 is that adding regulatory information to a GSMM gives robustness inferences that are likely closer to the underlying biological reality than is the GSMM alone. Thus, since the authors did not infer a general regulatory network, the higher robustness of the housekeeping genes is likely an artifact of the absence of regulatory information here. Overall, the manuscript makes clear that the authors are aware of this issue, but the wording is not as careful as it could be in a number of places.

See especially lines 223-232 and 281-284 for more instances of this issue.

We completely agree with the reviewer that adding regulatory information can only reduce robustness predicted by a GSMM. We also agree that GSMM represents only the full potential robustness of the metabolism but may not correspond to the true robustness when regulation is added. We tried discussing more explicitly this issue in the manuscript (see lines 246-248 and 400-406). We also agree that adding regulatory information may lead to inference of robustness closer to the real robustness of the organism. A key objective of our study was to assess the impact of a regulatory constraint, (*i.e* the VRN) on the robustness predicted by a GSMM. The reviewer mentioned that in many cases redundant enzymes do not complement each other due to different regulation. This is the point that we wanted to investigate, *i.e* in which conditions the VRN does not allow the regulated redundant enzymes to complement the first copy due to a lack of expression.

We also agree that adding the general regulatory network would be nice to infer its specific contribution to robustness and the putative crosstalk with the VRN. This aspect should be addressed in a future study. We recognize that our work is a first step toward an estimation of phenotypic robustness and do not describe the “real” (accurate) robustness of a biological system. Nevertheless, it provides a first evaluation on the variation of robustness in a pathogen due to the VRN regulation. We agree that the high robustness ($R=1$) of the housekeeping functions upon environmental

perturbations is likely an artifact due to the missing general regulatory network. However, the differential impact of the VRN on housekeeping versus virulence functions is not artifactual and results from differences in the VRN plugging into the metabolic network, i.e. connection at genes involved in genetic or functional redundancy on primary metabolism.

As stated above, we have added comments in two places in the manuscript to weight some of the statements on this aspect:

Lines 246-248: It cannot be ruled out however that some redundant genes involved in housekeeping functions may be also controlled by regulatory networks other than the VRN, and thus could become essential under specific environments.

Lines 400-406: Because the *R. solanacearum* global regulatory network (including metabolic regulation, stress response, etc...) is not yet reconstructed, the impact of this network could not be evaluated. However, this study revealed the direct effect of the VRN on phenotypic robustness since the analyses were conducted in a straight comparison between activated/inactivated VRN states independently of the global regulatory network; future investigations are required to undercover potential crosstalks between the *R. solanacearum* VRN and the global regulatory network.

31. On lines 257-261, the authors claim that the VRN changes estimated robustness most strongly for “proliferation.” But since for the cell to proliferate a very large number of pathways must function correctly, it would seem obvious that this phenotype would be the more sensitive than phenotypes impacting a smaller number of pathways. Now, I may well misunderstand what the authors mean here, but in that case, this section needs clarification.

We understand that this prediction could seem obvious. However, if the VRN had not controlled genes involved in proliferation phenotypes, the variation of robustness would have been null. We agree that the low robustness of proliferation upon internal perturbations is certainly due to the high number of pathways involved in this phenotype; however, the objective of this analysis was to assess the reduction of robustness due to the VRN. Thus, instead of the number of pathways contributing to the phenotypes, the important parameter to estimate the differential robustness (with or without VRN) is the number of VRN-regulated genes contributing to the phenotypes. A higher connection of the VRN into the primary metabolism impacting proliferation was therefore unforeseeable. We tried to clarify this point in the text, see lines 244-246.

32. Likewise, I almost could not make heads or tails of lines 262-275—this section needs heavy rewriting.

This section was entirely rewritten (lines 281-300) and we hope it is now more comprehensive.

33. I think one of the reasons I had difficulties with several of these sections is that the authors, in an attempt to make the manuscript accessible, have used common terms like “phenotype” and “environmental perturbation” to refer to particular features of their model. However, one has to dig deeply into the supplements to find the actual model definitions of these abstractions. The authors need to clarify and give examples: for instance, a phenotype might mean the ability to produce a certain compound on a certain media according to their model.

We agree with the reviewer’s point of view but it is a thin balance to make the manuscript accessible to a broad audience and to be accurate in describing the investigated features. We have mentioned the specific phenotypes in the figures and supplementary data; we also tried to provide explanatory figures (Figs 1 and 7) to help the reader with the definitions. The definitions of ‘Environmental perturbation’, ‘phenotype’ and some others are now provided explicitly in the manuscript, see lines 207-214.

34. Likewise, “environmental perturbation” calls to mind a quantitative change in the environment (lower pH, higher temperature, reduced concentration of some nutrient or hormone). However, GSMM are notorious for not being able to model such changes well. So many of the “perturbations” are going to be more like presence/absence tests—loss of a nutrient, gene deletion and so forth. The authors need to make the reader aware of this distinction.

We agree that GSMM are not able to model quantitative changes. In the case of our hybrid model, the reconstructed regulatory network takes into account the quantitative changes in the environment (see for instance Supp. Mat. Table S8), even if it converts the information in several discrete values. A finer modeling of perturbations (by ODE for instance) could be possible but it would be hardly suitable with genome scale. Furthermore, the parametrization of such a model would be difficult because we often lack biological knowledge about quantitative effects of environment changes. For these reasons, we think that our methodological choices are reasonable to cover the biological questions that we addressed in this paper.

35. Minor comments:

“Ortholog” is misused in Figure 6: either a gene is or is not an ortholog of a particular gene (e.g., had a common ancestor at the last common ancestral species or did not).

We changed the term ortholog by homolog.

We hope that this revised version will prove satisfactory.

REVIEWERS' COMMENTS:

Reviewer #1 (Remarks to the Author):

The revised manuscript addresses our concerns. We appreciate the detail with which the authors have responded to our concerns.

Reviewer #2 (Remarks to the Author):

The authors have responded in detail to all my comments and I'm satisfied with their answers and changes to the manuscript. I commend them on an excellent analysis.

Reviewer #3 (Remarks to the Author):

Re-review of Peyraud, Cottret, Marmiesse and Genin: "Control of primary metabolism by a virulence regulatory network promotes phenotypic robustness in a bacterial plant pathogen."

Comments:

The authors have largely addressed my concerns with the previous manuscript. At the same time, I still feel that the manuscript implies too much that the patterns of robustness modeled are biological and not methodological in origins. I believe an explicit statement that the original modeled robustness before the VRN was added was nonbiological and the VRN brings the model closer to that reality is preferable to a claim of "reduced robustness" with the addition of the VRN.

Minor comments:

The edited section from lines 281-300 needs proofing: e.g., "Materials and Methods" on line 284 and Chi-square test on line 289?

Manuscript NCOMMS-17-17194A – Response to Referee’s comments

Comments:

The authors have largely addressed my concerns with the previous manuscript. At the same time, I still feel that the manuscript implies too much that the patterns of robustness modeled are biological and not methodological in origins. I believe an explicit statement that the original modeled robustness before the VRN was added was nonbiological and the VRN brings the model closer to that reality is preferable to a claim of “reduced robustness” with the addition of the VRN.

In the revised version, the statement was not as explicit as requested by the referee but we already introduced comments in the discussion (lines 404-411) about the robustness before the VRN was added and the potential bias it may cause. We notably mentioned that the global regulatory network is missing and its addition would bring the system closer to the real (*i.e* biological) robustness.

In order to avoid ‘claiming’ that the VRN reduce robustness when added, we have modified in this second revision the sentence on line 404 as follows:

‘This explains why control exerted by the VRN could restrict in a large extent the phenotypic robustness of virulence functions with respect to environmental perturbations’.

Minor comments:

The edited section from lines 281-300 needs proofing: e.g., “Materials and Methods” on line 284 and Chi-square test on line 289?

This has been corrected.